# ATR inhibition facilitates targeting of leukemia dependence on convergent nucleotide biosynthetic pathways

Thuc M. Le[1,2], Soumya Poddar[1,2], Joseph R. Capri[1,2], Evan R. Abt[1,2], Woosuk Kim[1,2], Liu Wei[1,2], Nhu T. Uong[1,2], Chloe M. Cheng[1,2], Daniel Braas[1,3,4], Mina Nikanjam[5], Peter Rix[6], Daria Merkurjev[7], Jesse Zaretsky[1,4], Harley I. Kornblum[1,8,9,10], Antoni Ribas[1,4,5,9,11,12], Harvey R. Herschman[1,2,4,13], Julian Whitelegge[14], Kym F. Faull[1,2,14], Timothy R. Donahue[1,2,4,11], Johannes Czernin[1,2] & Caius G. Radu [1,2]

Leukemia cells rely on two nucleotide biosynthetic pathways, de novo and salvage, to produce dNTPs for DNA replication. Here, using metabolomic, proteomic, and phospho-proteomic approaches, we show that inhibition of the replication stress sensing kinase ataxia telangiectasia and Rad3-related protein (ATR) reduces the output of both de novo and salvage pathways by regulating the activity of their respective rate-limiting enzymes, ribonucleotide reductase (RNR) and deoxycytidine kinase (dCK), via distinct molecular mechanisms. Quantification of nucleotide biosynthesis in ATR-inhibited acute lymphoblastic leukemia (ALL) cells reveals substantial remaining de novo and salvage activities, and could not eliminate the disease in vivo. However, targeting these remaining activities with RNR and dCK inhibitors triggers lethal replication stress in vitro and long-term disease-free survival in mice with B-ALL, without detectable toxicity. Thus the functional interplay between alternative nucleotide biosynthetic routes and ATR provides therapeutic opportunities in leukemia and potentially other cancers.

[1] Department of Molecular and Medical Pharmacology, University of California, Los Angeles, Los Angeles, CA 90095, USA. [2] Ahmanson Translational Imaging Division, University of California, Los Angeles, Los Angeles, CA 90095, USA. [3] UCLA Metabolomic Center, University of California, Los Angeles, Los Angeles, CA 90095, USA. [4] David Geffen School of Medicine, University of California, Los Angeles, Los Angeles, CA 90095, USA. [5] Division of Hematology-Oncology, University of California, Los Angeles, Los Angeles, CA 90095, USA. [6] Vector Pharma Advisors Inc., San Diego, CA 92130, USA. [7] Department of Microbiology, Immunology & Molecular Genetics, University of California, Los Angeles, Los Angeles, CA 90095, USA. [8] Department of Psychiatry and Biobehavioral Sciences and Semel Institute for Neuroscience and Human Behavior, University of California, Los Angeles, Los Angeles, CA 90095, USA. [9] Jonsson Comprehensive Cancer Center, University of California, Los Angeles, Los Angeles, CA 90095, USA. [10] Eli and Edythe Broad Center of Regenerative Medicine and Stem Cell Research, University of California, Los Angeles, Los Angeles, CA 90095, USA. [11] Department of Surgery, University of California, Los Angeles, Los Angeles, CA 90095, USA. [12] Department of Medicine, Division of Surgical Oncology, University of California, Los Angeles, Los Angeles, CA 90095, USA. [13] Department of Biological Chemistry, University of California, Los Angeles, Los Angeles, CA 90095, USA. [14] The Pasarow Mass Spectrometry Laboratory, Neuropsychiatric Institute-Semel Institute for Neuroscience and Human Behavior, University of California, Los Angeles, Los Angeles, CA 90095, USA. Thuc M. Le, Soumya Poddar and Joseph R. Capri contributed equally to this work. Correspondence and requests for materials should be addressed to C.G.R. (email: cradu@mednet.ucla.edu)

Unabated proliferation is a hallmark of cancer which requires new DNA synthesis from deoxyribonucleotide triphosphates (dNTPs). However, cellular dNTP levels only suffice to sustain a few minutes of DNA replication indicating that dNTP pools are produced 'on demand' via tightly regulated biosynthetic pathways[1]. These deoxynucleotide biosynthetic pathways, termed de novo and salvage, rely on distinct carbon and nitrogen sources[2]. De novo pathways use glucose and amino acids to produce ribonucleotide diphosphates (rNDPs) which are converted into deoxyribonucleotide diphosphates (dNDPs) by ribonucleotide reductase (RNR), a two-subunit enzyme complex[3] upregulated in most cancers[4]. Salvage pathways convert preformed ribonucleosides, deoxyribonucleosides and nucleobases into nucleotides through the actions of metabolic kinases and phosphoribosyltransferases[2]. Amongst nucleoside salvage kinases, deoxycytidine kinase (dCK) has the broadest substrate specificity, encompassing both purine and pyrimidine nucleosides[5]. While tumors are thought to predominantly rely on de novo pathways to produce nucleotides[6], scavenging of preformed nucleosides via dCK and other salvage kinases may also play important roles in the economy of nucleotide metabolism in cancer cells. Many of the cell lines included in the Cancer Cell Line Encyclopedia[7, 8] express dCK at higher levels than the corresponding normal tissues. Increased tumor dCK expression relative to matched normal tissues also occurs in patient samples, as evidenced by RNASeq data from The Cancer Genome Atlas (TCGA, http://cancergenome.nih.gov)[9, 10]. Moreover, in vivo, cancer cells often encounter limited supplies of essential de novo pathway substrates, e.g., glucose, glutamine and aspartate, because of their avid consumption of these nutrients and inadequate vascularization[11]. An insufficient de novo biosynthetic capacity, coupled with an increased demand for dNTPs due to unabated proliferation, might increase the dependency of certain tumors on salvage pathways for nucleotide production. Consistently, we previously showed that acute lymphoblastic leukemia (ALL) cells display nucleotide biosynthetic plasticity[12], defined as the ability to compensate for the inhibition of either de novo or salvage pathways by upregulating the alternate pathway. These metabolic transitions occurred both in vitro and in vivo; moreover, partial inhibition of both de novo and salvage biosynthetic routes was required for therapeutic activity in animal models of T and B-ALL[12].

Collectively, these results suggest that, in acute leukemia, and potentially in other cancers, nucleoside salvage biosynthetic pathways may be metabolic non-oncogene addictions[13] targetable by specific inhibitors. However, since both de novo and salvage biosynthetic pathways also operate in normal cells[14, 15], a better understanding of the signaling mechanisms that regulate their activity in cancer cells may lead to the development of more effective targeted therapies. In this context, the mTOR[16–18], Myc[19, 20] and Ras[21] pathways have been shown to regulate nucleotide biosynthesis. The replication stress response pathway also plays important roles in regulating nucleotide metabolism, given its unique ability to 'sense' dNTP insufficiency[22]. The most proximal enzyme in the cellular response to replication stress is ataxia telangiectasia and Rad3-related protein (ATR), a serine threonine kinase activated at stalled replication forks[23] in response to nucleotide insufficiency and other replication defects. In addition to its well-established role in regulating origin firing and promoting fork stability[24], ATR has been recently linked to nucleotide metabolism. Inhibition of ATR, or of its downstream effector kinases CHEK1 and WEE1, reduces dNTP levels in cancer cell lines[25]. This effect of ATR inhibition was proposed to involve the downregulation of the small RNR subunit RRM2, particularly at the G1/S transition[26, 27]. Intriguingly, ATR also regulates dCK activity in several solid tumor and myeloid leukemia cells by phosphorylation at serine 74[28]). This post-translational modification (PTM) modulates dCK's catalytic properties and substrate specificity[29, 30]. While collectively these findings support a connection between ATR signaling and dNTP production, the metabolic consequences of ATR inhibition in malignancies with nucleotide biosynthetic plasticity are yet to be defined.

Here, we examine ATR modulation of dNTP synthesis and utilization for DNA synthesis, and the consequences for tumor cell viability in culture and in vivo in ALL models, using quantitative approaches. Our targeted multiplexed mass spectrometric (MS) assay measures the differential contributions of the de novo and salvage pathways both to nucleotide pools and newly replicated DNA. This assay is used in conjunction with proteomic and phosphoproteomic MS approaches to investigate the mechanisms responsible for alterations in nucleotide biosynthesis induced by ATR inhibition. In addition, we compare direct targeting of de novo and salvage rate-limiting enzymes, using specific inhibitors vs. indirect inhibition of these enzymes via interference with ATR signaling. These studies identify a synthetically lethal interaction between inhibition of convergent nucleotide biosynthetic routes and ATR in ALL. This combination is therapeutically exploitable in vivo, resulting in long-term, disease-free survival in a systemic $p185^{BCR-ABL}Arf^{-/-}$ pre-B-ALL mouse model representative of the human disease[31–33]. Overall, our findings suggest that nucleotide biosynthetic plasticity in lymphoblastic leukemia cells, and potentially in other malignancies, is mediated by both ATR signaling and nucleotide metabolic adaptive mechanisms which may be targetable without overt toxicity to normal tissues, using existing small molecule inhibitors.

## Results

**ATR and dCK Co-inhibition impairs G1/S transition.** Human T-ALL cells CCRF-CEM (CEM) express dCK and exhibit constitutive phosphorylation of the ATR effector kinase CHEK1 on Serine 345 (pS345, Supplementary Fig. 1a), a marker of replication stress[34]. CHEK1 pS345 levels are reduced following exposure to VE-822, a specific ATR inhibitor[35] (Supplementary Fig. 1a). To investigate whether ATR inhibition increases the dependence of T-ALL cells on dCK activity at the G1/S transition, CEM cells were synchronized in G1 using Palbociclib, a CDK4/6 inhibitor[36, 37], and then released into media containing VE-822 and/or DI-82, a high-affinity dCK inhibitor (dCKi) developed by our group[38]. At various time points following G1 release, cells were pulsed for 1 h with 5'-ethynyl-2'-deoxyuridine (EdU) to analyze cell cycle kinetics by flow cytometry. Six hours after release from G1, ~ 25% of cells in the untreated and single drug treated groups advanced into early S-phase (designated as S1, in blue, Fig. 1a). In contrast, only 16% of cells treated with both VE-822 and dCKi entered S1. Twelve hours after release from G1 arrest, 16% fewer VE-822-treated cells entered the later part of S-phase (designated as S2, in red, Fig. 1b) compared to untreated cells. While at this time point dCK inhibition alone did not affect the number of cells that progressed beyond early S phase, progression to late S-phase was significantly impeded (Supplementary Fig. 1b). Co-inhibition of ATR and dCK decreased the percentage of cells that reached S2 by fivefold relative to untreated cells (Fig. 1b). The effects of ATR inhibition on cell cycle kinetics were partially rescued by nucleotide supplementation, in a dCK-dependent manner (Supplementary Fig. 1c, d).

To further investigate the functions of dCK and ATR at the G1/S transition in CEM cells, a non-targeted liquid chromatography mass spectrometry (LC-MS) assay was used to determine the utilization of labeled [$^{13}C_6$]glucose and [$^{13}C_9,^{15}N_3$]deoxycytidine, the main substrates for de novo and salvage nucleotide

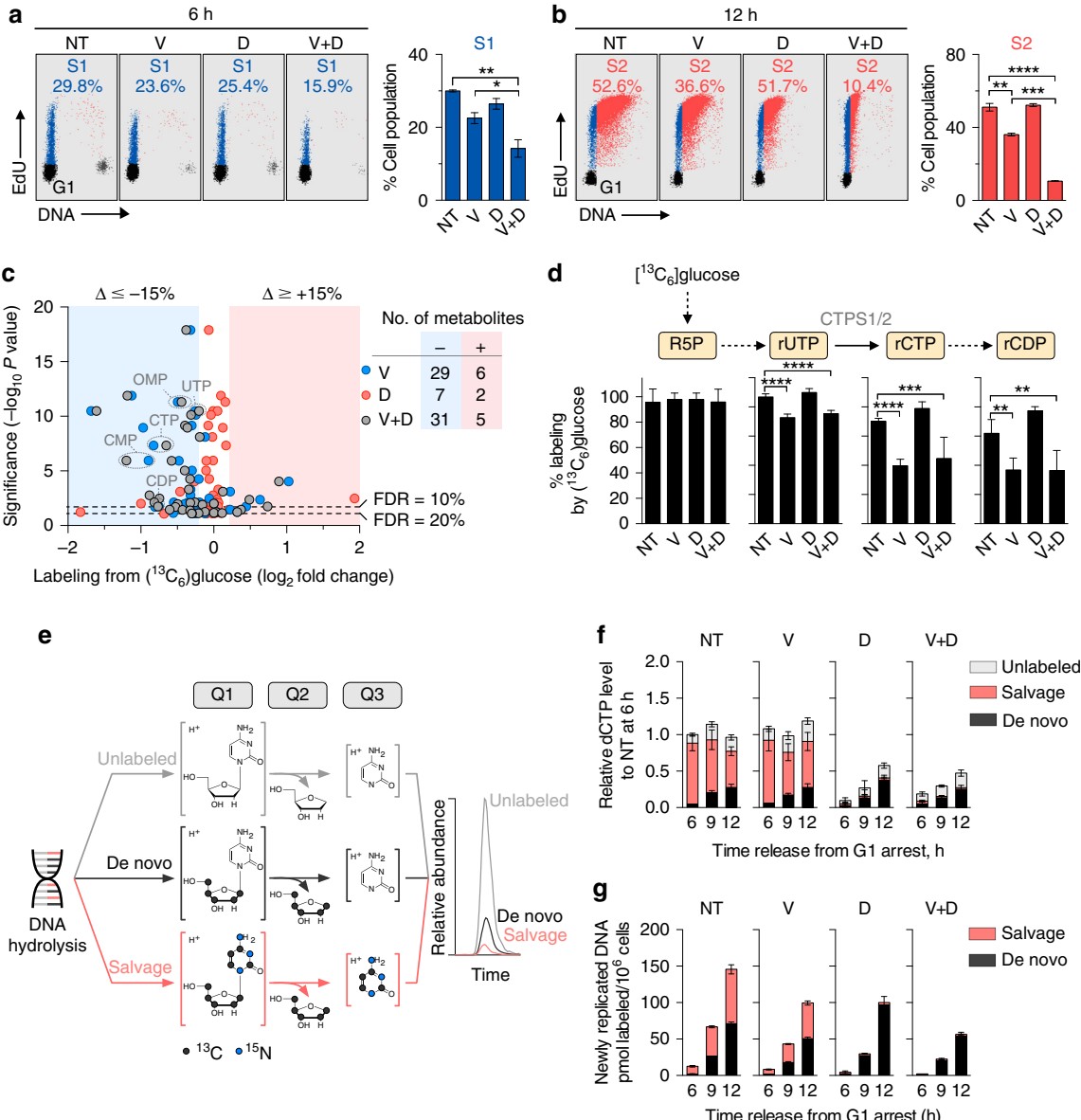

**Fig. 1** Effects of ATR and dCK inhibition on G1-S transition and substrate utilization for dCTP biosynthesis. **a**, **b** Flow cytometry analysis of EdU incorporation in CEM T-ALL cells treated with VE-822 (1 μM) and/or dCKi (DI-82,1 μM) for 6 **a** and 12 h **b** following release from G1 arrest, respectively. Bar graphs summarize the percentage of cell populations in S1 (early S-phase) and S2 (mid to late S-phase) at 6 and 12 h (mean ± s.d., n = 2, one-way ANOVA, Bonferroni corrected). Plots are representative of two independent experiments. **c** Comparison of metabolite labeling by [$^{13}C_6$]glucose in CEM T-ALL cells treated with VE-822 and/or dCKi for 12 h following release from G1 arrest. Number of metabolites exhibiting alterations in [$^{13}C_6$]glucose labeling greater than 15% with significance at a false discovery rate ≤ 20% are indicated. **d** Percent glucose labeling of ribonucleotides intermediates in the de novo dCTP biosynthesis (mean ± s.d., n = 6, one-way ANOVA, Bonferroni corrected). **e** Workflow for targeted LC-MS/MS-MRM analysis of dCTP incorporated into newly replicated DNA using a triple quadrupole mass spectrometer. See text for details and Supplementary Fig. 4 for the LC-MS/MS-MRM analysis of dCTP pools. **f**, **g** Contributions of the de novo and salvage pathways to dCTP pools **f** and dCTP incorporated into newly synthesized DNA **g** in CEM cells treated with VE-822 and/or dCKi after release from G1 arrest (mean ± s.d., n = 3). Results are representative of two independent experiments. NT = Not treated, V = VE-822, D = dCKi, V + D = VE-822 + dCKi. *P < 0.05; **P < 0.01; ***P < 0.001; ****P < 0.0001. *ATR* ataxia telangiectasia and Rad3-related protein, *EDU* 5'-ethynyl-2'-deoxyuridine, *FDR* false discovery rate, *RNR* ribonucleotide reductase, *CTPS1/2* CTP synthase 1/2, *R5P* ribose 5-phosphate, *OMP* orotidine monophosphate, *rCDP* cytidine diphosphate, *CTP* cytidine triphosphate, *UTP* uridine triphosphate, *dCTP* deoxycytidine triphosphate, *LC-MS/MS-MRM* liquid chromatography tandem mass spectrometry operating in multiple reaction monitoring

biosynthesis, respectively. Of the 166 metabolites identified in CEM cells treated with VE-822 and/or dCKi, 105 metabolites found in all four treatment groups contained glucose-derived $^{13}C$ atoms. While ATR inhibition did not decrease glucose uptake and labeling of glycolytic intermediates (Supplementary Fig. 2), it significantly decreased glucose utilization for 29 other metabolites (Fig. 1c and Supplementary Data 1). These metabolites included

intermediates such as rUTP, rCTP and rCDP in the de novo dCTP biosynthesis (Fig. 1d). These data indicate that ATR inhibition impacts glucose utilization for de novo nucleotide biosynthesis. However, several deoxyribonucleotides, including dCTP, were below the limit of detection of the non-targeted LC-MS approach, raising the concern that the sensitivity of this assay is not sufficient to measure the contribution of the salvage

**Table 1 Rates of de novo and salvage produced dCTP incorporation into newly replicated DNA**

| Treatment | De novo[a] | Salvage[a] | Ratio (salvage/de novo) |
|---|---|---|---|
| Not treated | 0.20 ± 0.01 | 0.17 ± 0.01 | 0.83 |
| VE-822 | 0.17 ± 0.01 | 0.14 ± 0.01 | 0.85 |
| dCKi | 0.23 ± 0.01 | 0.019 ± 0.004 | 0.082 |
| VE-822 + dCKi | 0.19 ± 0.01 | 0.022 ± 0.002 | 0.12 |

[a]Slope ± standard error of regression line in arbitrary unit

pathway or the ratio of de novo and salvage biosynthesis to these pools. To address this problem, a newly developed targeted MS assay was developed (Fig. 1e and Supplementary Fig. 3). In this assay, samples containing either extracted dNTPs or hydrolyzed DNA from labeled cells are separated by LC for detection by a triple quadrupole mass spectrometer using multiple reaction monitoring (LC-MS/MS-MRM). The first (Q1) and third (Q3) quadrupoles function as mass filters, while the second (Q2) quadrupole serves as a collision chamber (Fig. 1e). For instance, to profile the biosynthetic composition of deoxycytidine (dC) derived from hydrolyzed DNA or dCTP, an intact, protonated dC ion is selected in Q1, followed by fragmentation of the glycosidic bond, which releases protonated cytosine in Q2, that is filtered in Q3 and detected to generate an ion chromatogram. The peak areas for the ion chromatograms of salvage $[^{13}C_9,^{15}N_3]$dC (red trace), de novo $[^{13}C_5]$dC (black trace, from $[^{13}C_6]$glucose) and unlabeled dC (gray trace) (Fig. 1e) are used to determine the relative contributions of the de novo and salvage routes both to dCTP pools and to DNA-incorporated dCTP.

Since dCK phosphorylates not only dC but also dA and dG[5], the targeted LC-MS/MS-MRM assay was used to determine whether, in T-ALL cells, dCK mediates the salvage of multiple deoxyribonucleosides (dNs). However, salvaging of purine dNs via dCK occurred only if the catabolic enzymes which degrade these dNs, adenosine deaminase (ADA) and purine nucleoside phosphorylase (PNP), were inhibited pharmacologically (Supplementary Figs 4 and 5). Since inactivating mutations in ADA and PNP have been associated with severe combined immunodeficiency[39], but not with cancer, we focused on examining the metabolic fate of dC as the most relevant dCK substrate in ATR-inhibited T-ALL cells for salvage biosynthesis. CEM cells were collected at multiple points after release from Palbociclib-induced G1 arrest into media containing VE-822 and/or dCKi, as well as substrates for the de novo and salvage pathways, $[^{13}C_6]$glucose and $[^{13}C_9, ^{15}N_3]$dC, respectively. At each of the examined time points, free dCTP pools in untreated cells were predominantly synthesized by the salvage pathway from $[^{13}C_9,^{15}N_3]$dC via dCK, with only a small contribution from $[^{13}C_6]$glucose via the de novo pathway (Fig. 1f). While ATR inhibition alone did not alter free dCTP levels or their biosynthetic origins (e.g., de novo vs. salvage), dCKi either alone or in combination with VE-822, nearly eliminated the contribution of the salvage pathway and reduced the amount of free dCTP by ~ 50% at the 12 h time point.

In contrast to the free dCTP pool, which was predominantly derived from the salvage pathway, dCTP incorporated into newly replicated DNA of CEM cells was produced in equal proportions by the de novo and salvage pathways (Fig. 1g and Table 1). This observation is consistent with previous findings that de novo synthesized dCTP is more readily incorporated in DNA than is dCTP synthesized by the salvage pathway[12]. ATR inhibition reduced the DNA incorporation of both de novo and salvage produced dCTP, yielding a combined 30% reduction in overall DNA labeling compared to untreated cells at the 12 h time point (Fig. 1g and Table 1). This reduction is consistent with data in the

literature showing replication fork collapse and delays in restarting DNA replication following ATR inhibition[40–43]. dCK inhibition abolished the incorporation of salvage produced dCTP into DNA and triggered a compensatory increase in the DNA incorporation of de novo generated dCTP. This compensatory response was suppressed in cells treated with both VE-822 and dCKi (Fig. 1g and Table 1).

**Effects of ATR inhibition on RRM2 and dCK.** To investigate the molecular mechanisms underlying the metabolic consequences of ATR and dCK inhibition, global changes in protein expression were assessed in CEM cells using quantitative nano-LC tandem MS (nLC-MS/MS) (Fig. 2a). Chemical isotope coding following reductive dimethylation of peptide N-termini and lysine primary amines with differential stable isotopes (light:medium:heavy) was used to compare expression of experimental (VE-822, dCKi and VE-822 + dCKi) samples to controls (untreated, NT) by mixing equal proportions for triplex nLC-MS/MS quantitative analyses. The data set was filtered for proteins identified in all treatment groups in three independent experiments with coefficients of variation <20%. This yielded 1757 proteins with relative fold changes in treated vs. untreated cells ranging from 0.45 to 1.83. Of these, ~3% (46 proteins) displayed statistically significant (as determined by one-way ANOVA and false discovery rate cutoffs) fold changes in expression (>20%) in at least one treatment group (Fig. 2b and Supplementary Data 2). Protein levels of both RNR subunits, RRM1 and RRM2, as well as thymidylate synthase decreased by more than 20% following ATR inhibition (Fig. 2c). Changes in the expression of de novo enzymes observed in synchronous cells also occurred with ATR inhibition in asynchronous CEM cells (Fig. 2d and Supplementary Data 3), thereby arguing against the possibility of experimental artifacts introduced by Palbociclib-mediated cell cycle synchronization. The reduction in RRM2 levels induced by ATR inhibition was accompanied by an ~ 50% decrease in the phosphorylation of RRM2 on threonine 33 (pT33) (Fig. 2e), a phosphosite previously linked to the RRM2 stability[44].

ATR inhibition decreased dNTP levels in several solid tumor-derived cell lines and has been linked to reduced RRM2 levels[26, 45]. However, it remains unclear to what degree RRM2 protein levels are rate-limiting for de novo dCTP biosynthesis. To further investigate the relationship between RRM2 protein levels and dCTP biosynthesis, we knocked down RRM2 in CEM cells using shRNA (Supplementary Fig. 6a). RRM2 levels in the CEM shRNA$^{RRM2}$ cells were reduced by 35–50%, as determined by quantitative nLC-MS/MS and intracellular flow cytometry analyses (Supplementary Fig. 6a, b). CEM shRNA$^{RRM2}$ cells exhibited ~30% lower incorporation of de novo synthesized dCTP into newly replicated DNA compared to control cells (Supplementary Fig. 6c); a response comparable with the effects of pharmacological ATR inhibition (Fig. 1g). These findings suggest that the RRM2 regulation by ATR is an important determinant of de novo dCTP biosynthesis in T-ALL cells. Since ATR inhibition reduced RRM2 levels by only 20%, it is likely that there are

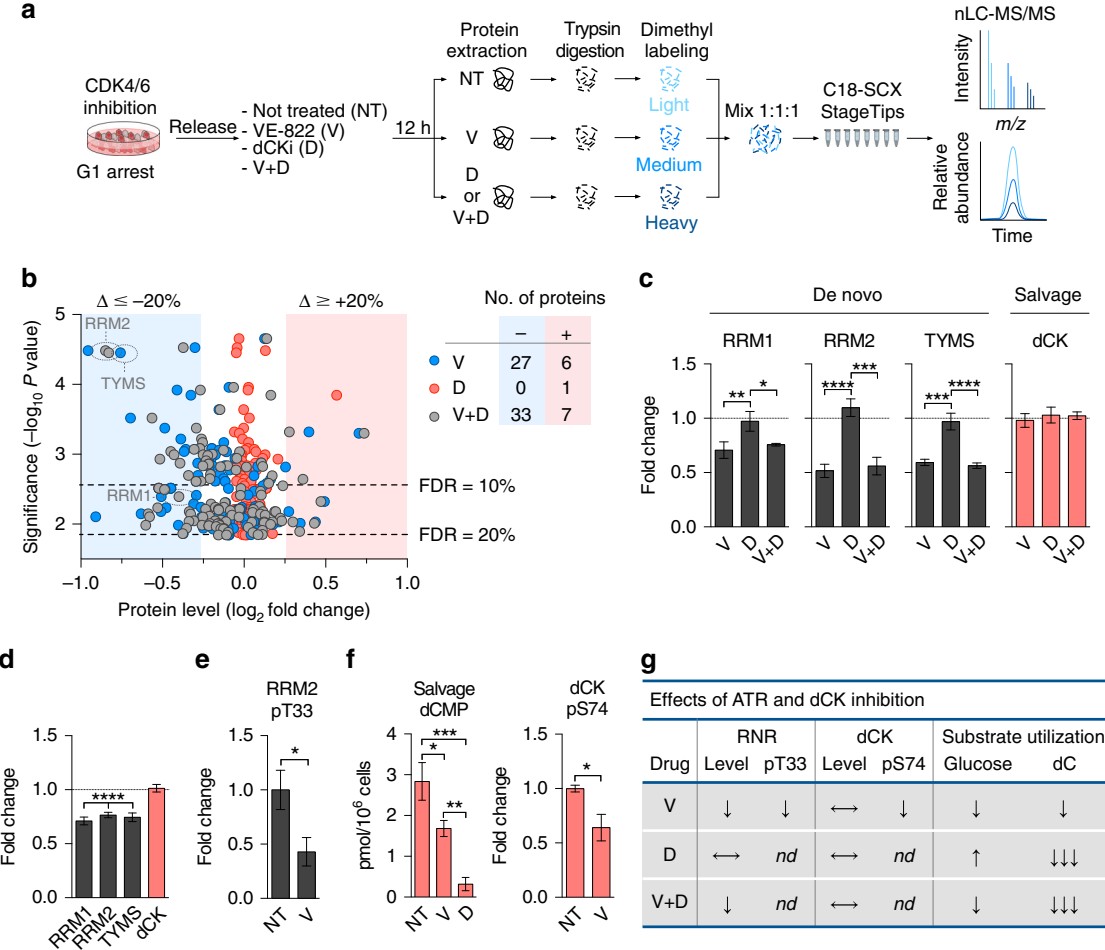

**Fig. 2** Alterations in total protein and phosphoprotein levels following ATR and dCK inhibition. **a** Workflow for quantitative global proteomics using nLC-MS/MS. See text for details. **b** Comparison of protein levels in CEM cells treated with VE-822 and/or dCKi for 12 h following release from G1 arrest. Number of proteins exhibiting fold changes greater than 15% changes with significance at a false discovery rate ≤ 20% are indicated. **c** Protein levels of nucleotide biosynthetic enzymes (mean ± s.d., n = 3, one-way analysis of variance (ANOVA, Bonferroni corrected). **d** Protein levels in asynchronous CEM cells treated with VE-822 (1 μM) for 12 h (mean ± s.d., n = 3, one sample t-test to assess if the mean of the protein level normalized to untreated control is equal to one). **e** Relative level of RRM2 pT33 normalized to RRM2 protein level from **d**, in asynchronous CEM cells treated with VE-822 (1 μM) for 12 h (mean ± s.d., n = 3, unpaired two-tailed Student's t-test). (**f**, *left panel*) Salvage produced [$^{13}C_9$,$^{15}N_3$]dCMP in asynchronous CEM cells treated with VE-822 or dCKi for 12 h (mean ± s.d., n = 3, one-way ANOVA, Bonferroni corrected). (**f**, *right panel*) Relative levels of dCK pS74, after normalized to dCK protein level from **d**, in asynchronous CEM cells treated with VE-822 (1 μM) for 12 h (mean ± s.d., n = 3, unpaired two-tailed Student's t-test). **g** Summary of the observed effects of ATR and dCK inhibition in CEM cells. ↓ partial decrease/inhibition, ↓↓↓ nearly complete inhibition, ↑ increase, ⟷ no change, *nd* not determined. NT = Not treated, V = VE-822, D = dCKi, V + D = VE-822 + dCKi. *P < 0.05; **P < 0.01; ***P < 0.001; ****P < 0.0001. *nLC-MS/MS* nano liquid chromatography tandem mass spectrometry, *RRM1* ribonucleotide reductase subunit 1, *RRM2* ribonucleotide reductase subunit 2, *TYMS* thymidylate synthase, *dCK* deoxycytidine kinase, *dCMP* deoxycytidine monophosphate

other mechanisms by which ATR regulates de novo dCTP biosynthesis. These additional mechanisms could include reduced levels of the large RNR subunit, RRM1 (Figs. 2c, d), and/or changes in yet to be identified regulatory PTMs in RRM1 and RRM2 that are modulated directly or indirectly by ATR signaling.

In contrast to the reduced RRM1 and RRM2 levels in response to ATR inhibition, dCK protein levels were not affected (Fig. 2c, d). ATR was shown to directly phosphorylate dCK on serine 74 (dCK pS74) to control its activity under replication stress[28]. We therefore quantified the effects of ATR inhibition on the biosynthetic output and phosphorylation status of dCK. Used as a positive control, the dCKi reduced dCK activity, as defined by contribution of labeled [$^{13}C_9$,$^{15}N_3$]dC to intracellular dCMP, by ~ 90% (Fig. 2f, *left panel*). ATR inhibition in CEM cells reduced the dCK-labeled [$^{13}C_9$,$^{15}N_3$]dCMP pool by ~ 33% compared to untreated cells (Fig. 2f, *left panel*). The reduction in dCMP

biosynthesis following ATR inhibition correlated with a ~ 36% decrease in dCK pS74 levels (Fig. 2f, *right panel*). Collectively, these data (summarized in Fig. 2g) show that, in T-ALL cells, ATR regulates de novo and salvage pathways by diverse mechanisms involving alterations in total protein (ATR) and protein phosphorylation (dCK) levels. Nonetheless, both de novo and salvage pathways retain significant activity in ATR inhibited CEM cells.

**Increased salvage dCTP biosynthesis by RNR inhibition.** To identify the most potent clinically relevant RNR inhibitors that could be used to target the remaining de novo nucleotide biosynthetic activity in ATR inhibited CEM cells we evaluated four compounds, each with a distinct mechanism of action: (3-AP)[46, 47], hydroxyurea (HU)[48], gallium maltolate (GaM)[49] and thymidine (dT)[50] (Fig. 3a). Amongst these, 3-AP was the most

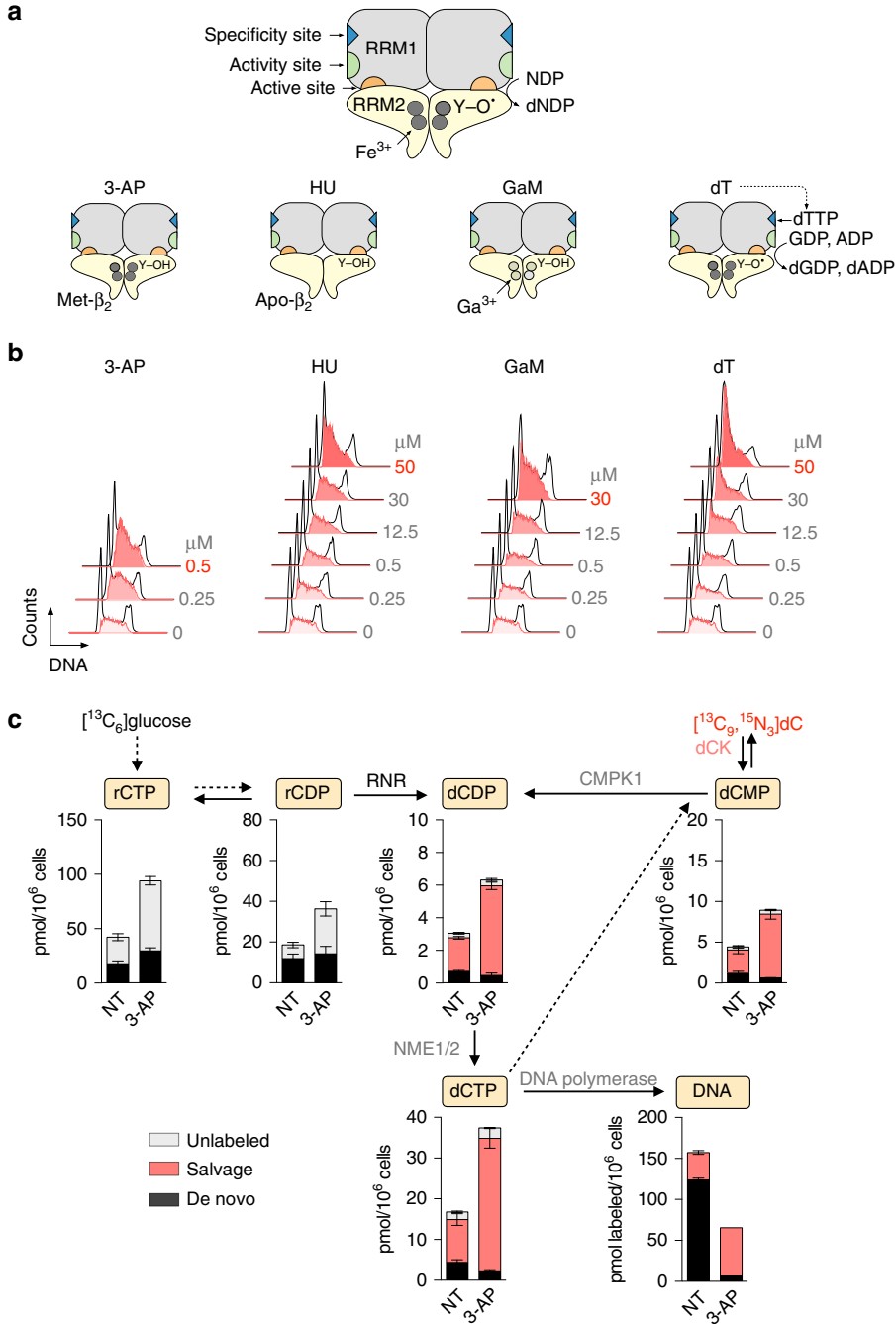

**Fig. 3** 3-AP potently inhibits RNR and enhances salvage nucleotide biosynthesis. **a** Mechanisms of action of four RNR inhibitors. The two RNR subunits, RRM1 (α) and RRM2 (β) form a catalytically active α₂β₂ complex. Each RRM1 subunit contains two allosteric regulatory sites (the specificity and activity sites), as well as the active site, where nucleotide reduction occurs. The active form of the RRM2 dimer (holo-β₂) houses the di-iron cofactor and the tyrosyl radical (Y-O•). 3-AP forms a complex with $Fe^{2+}$ which interferes with the regeneration of the tyrosyl radical in RRM2 therefore promoting the formation of an inactive met-β small subunit which retains its di-iron center[7]. Hydroxyurea (HU) scavenges the RRM2 tyrosyl radical and depletes the di-iron center to form an inactive apo-β form. Gallium maltolate (GaM) releases $Ga^{3+}$ which mimics $Fe^{3+}$ and disrupts the RRM2 di-iron center. Thymidine (dT) is converted via the salvage pathway to thymidine triphosphate (dTTP) which binds to the allosteric specificity site on RRM1 to favor GDP reduction over pyrimidine (CDP and UDP) reduction, thereby resulting in dCTP insufficiency. **b** Effects of RNR inhibitors on cell cycle progression. CEM cells were incubated for 24 h with indicated concentrations of RNR inhibitors followed by cell cycle analyses using flow cytometry. Shown in bold red are the concentrations of each RNR inhibitor required to induce a greater than 45% increase in the S-phase population, indicative of S-phase arrest due to nucleotide insufficiency. Cell cycle plots are representative of two independent experiments. See Supplementary Fig. 8 for quantification. **c** LC-MS/MS-MRM analysis of dCTP biosynthesis in CEM cells treated with 500 nM 3-AP for 12 h (mean ± s.d., n = 3). *NT* not treated. *CMPK1* uridine-cytidine monophosphate kinase 1, *NME1/2* nucleoside diphosphate kinase 1/2, *dC* 2′-deoxycytidine, *dCMP* deoxycytidine monophosphate, *dCDP* deoxycytidine diphosphate

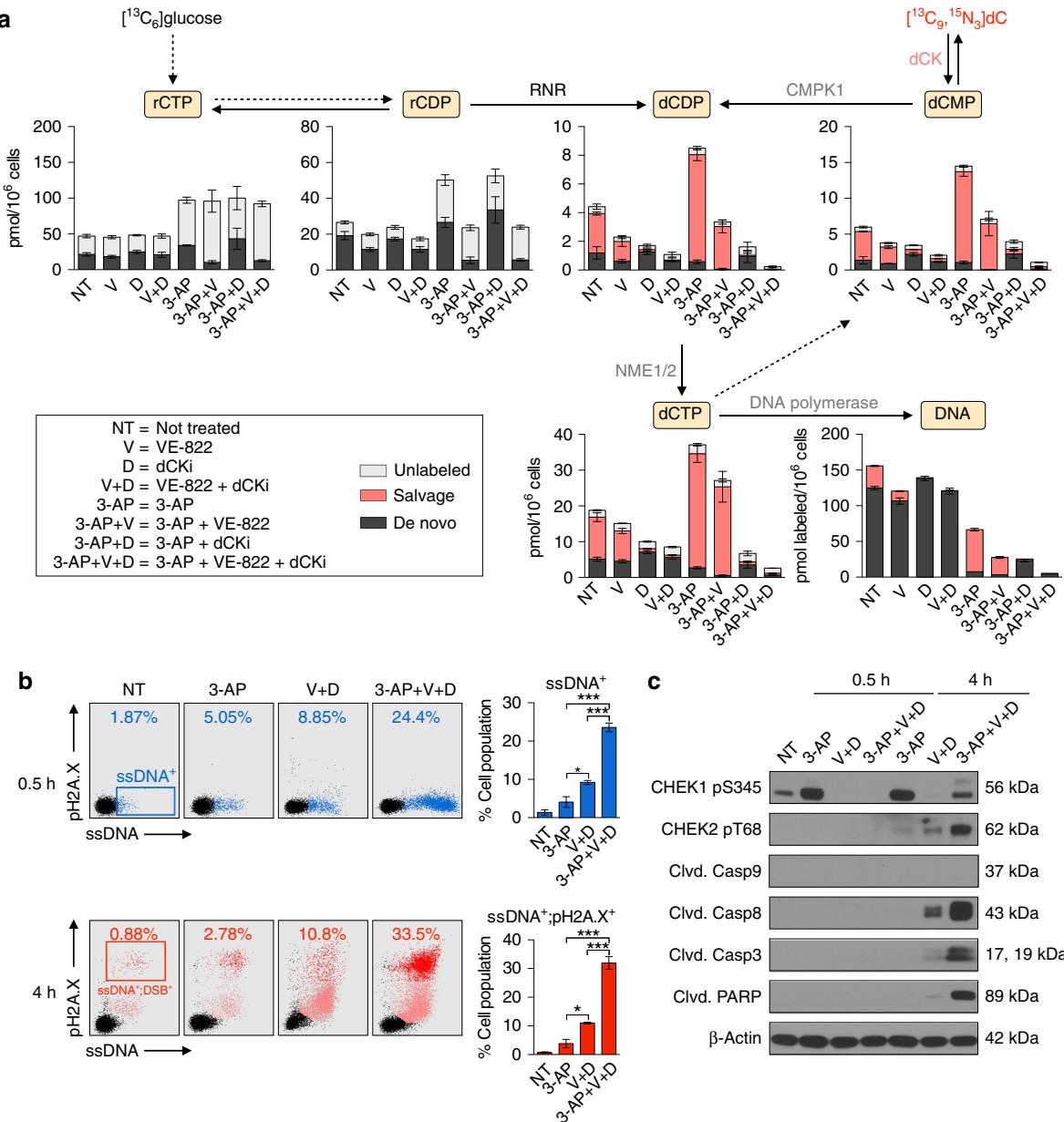

**Fig. 4** Synthetic lethality induced by combined inhibition of ATR, dCK and RNR. **a** LC-MS/MS-MRM analysis of dCTP biosynthesis in CEM cells treated as indicated in the text for 12 h (mean ± s.d., $n = 3$). Results are representative of two independent experiments. **b**, *left* Flow cytometry analyses of ssDNA (F7-26) and pH2A.X levels in CEM cells treated as indicated for 0.5 and 4 h. **b**, *right* Bar graphs summarizing the percentage of ssDNA⁺ and ssDNA⁺;pH2A. X⁺ cells at 0.5 and 4 h, respectively (mean ± s.d., $n = 2$, one-way ANOVA, Bonferroni corrected). ssDNA-pH2A.X plots are representative two independent experiments. **c** Representative immunoblots of CEM cells treated as indicated in the text for 0.5 and 4 h. *$P < 0.05$; **$P < 0.01$; ***$P < 0.001$; ****$P < 0.0001$. *ssDNA* single-stranded DNA, *DSB* double-stranded breaks, *PARP* Poly (ADP-ribose) polymerase

potent, as indicated by induction of S-phase arrest at concentrations as low as 0.5 µM (Fig. 3b and Supplementary Fig. 7). In contrast, 60–100-fold higher concentrations of HU, GaM and dT were required to induce S-phase arrest. The effects of 3-AP on the utilization of [¹³C₆]glucose and [¹³C₅, ¹⁵N₃]dC for nucleotide biosynthesis in CEM cells were investigated (Fig. 3c), using the targeted LC-MS/MS-MRM assay. 3-AP doubled the rCTP and rCDP pools, likely reflecting an inefficient conversion of these pools to dCDP via RNR. However, the most significant change in 3-AP treated cells was a ~ 19-fold reduction in the incorporation of de novo produced dCTP into DNA. Along with its effects on de novo biosynthesis, 3-AP triggered a compensatory upregulation of the salvage pathway. Salvage dC nucleotide pools doubled in size following 3-AP treatment, and the

incorporation of salvage-produced dCTP into DNA increased by >1.5-fold, thereby providing a potential mechanism of resistance to RNR inhibition by 3-AP (Fig. 3c).

**Triple combination induces lethal replication stress in vitro.** We next quantified the impact of combined ATR, dCK and RNR inhibition on the de novo and salvage dCTP biosynthesis in asynchronous CEM cells (Fig. 4a and Supplementary Data 4). ATR inhibition decreased the [¹³C₆]glucose labeling of the rCDP pool by 40% (Fig. 4a, rCDP panel), an effect similar to that observed in the synchronous model (Fig. 1d). In contrast, RNR inhibition increased the size of the rCDP pool (Fig. 4a, rCDP panel, 3-AP). While neither ATR nor RNR inhibition alone had a

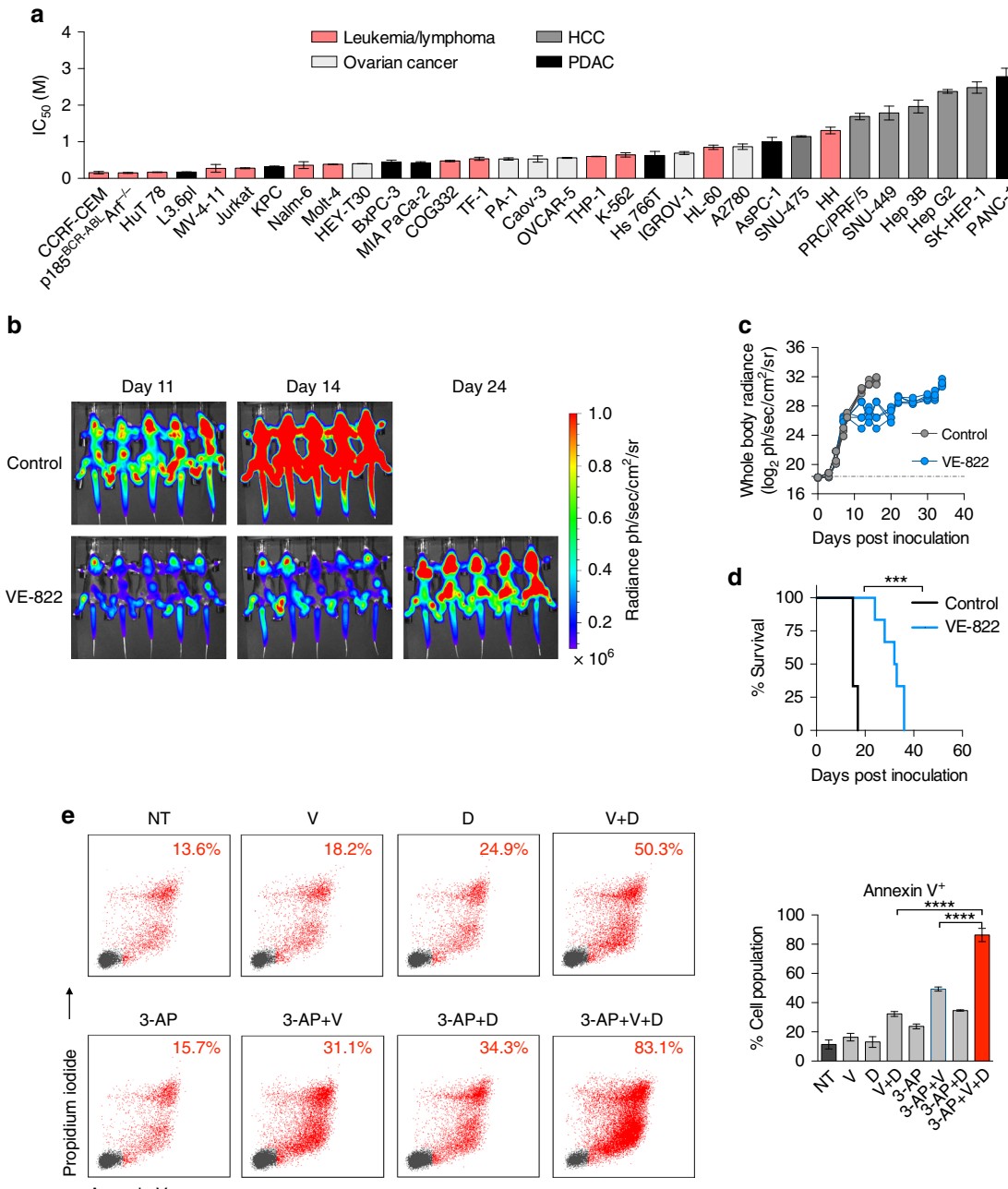

**Fig. 5** ATR inhibition alone is effective but not sufficient to achieve disease-free survival in a systemic primary B-ALL model. **a** IC$_{50}$ values of VE-822 in a panel of cancer cell lines and patient-derived samples (CellTiter-Glo assay at 72 h, mean ± s.d., $n = 3$). **b**, **c** Bioluminescence images **b** and quantification of whole-body radiance **c** of leukemia bearing mice treated with 40 mg kg$^{-1}$ VE-822 ($n = 6$) or vehicle (control, $n = 6$). VE-822 was administered once daily. **d** Kaplan-Meier survival analysis of C57BL/6 mice bearing p185$^{BCR-ABL}$Arf$^{-/-}$ systemic pre-B-ALL treated with 40 mg kg$^{-1}$ day$^{-1}$ VE-822 ($n = 6$) or vehicle (control, $n = 6$). Median survival for the control group was 15 days after treatment initiation and 32.5 days for the VE-822 group (Mantel–Cox test). **e** Apoptosis induction in p185$^{BCR-ABL}$Arf$^{-/-}$ pre-B-ALL cells treated as indicated (350 nM 3-AP, 100 nM VE-822, and 1 μM dCKi) for 72 h using flow cytometry for Annexin V and PI staining (mean ± s.d., $n = 2$, one-way analysis of variance, Bonferroni corrected). *$P < 0.05$; **$P < 0.01$; ***$P < 0.001$; ****$P < 0.0001$. *HCC* hepatocellular carcinoma, *PDAC* pancreatic ductal adenocarcinoma

statistically significant impact on the de novo contributions to the dCDP, dCMP and dCTP pools, these pools were nearly abolished when both ATR and RNR were inhibited simultaneously. However, the salvage biosynthetic contributions to the dCMP, dCDP and dCTP pools remained substantial in the absence of the dCKi. In fact, RNR inhibition, alone or combined with ATR inhibition, increased the salvage contributions to the dCMP, dCDP and dCTP pools by approximately twofold, indicative of a compensatory mechanism. dCK inhibition abolished this

adaptive mechanism to augment these deoxyribonucleotide pools. Consequently, the rate of dCTP incorporation into DNA was lowest when all three enzymes were inhibited (Fig. 4a, DNA panel).

Persistent nucleotide insufficiency triggers replication stress characterized by the accumulation of single-stranded DNA (ssDNA) at stalled replication forks, followed by DNA double-stranded breaks[51]. To investigate these events we used flow cytometry and antibodies against ssDNA[52] as an indicator of

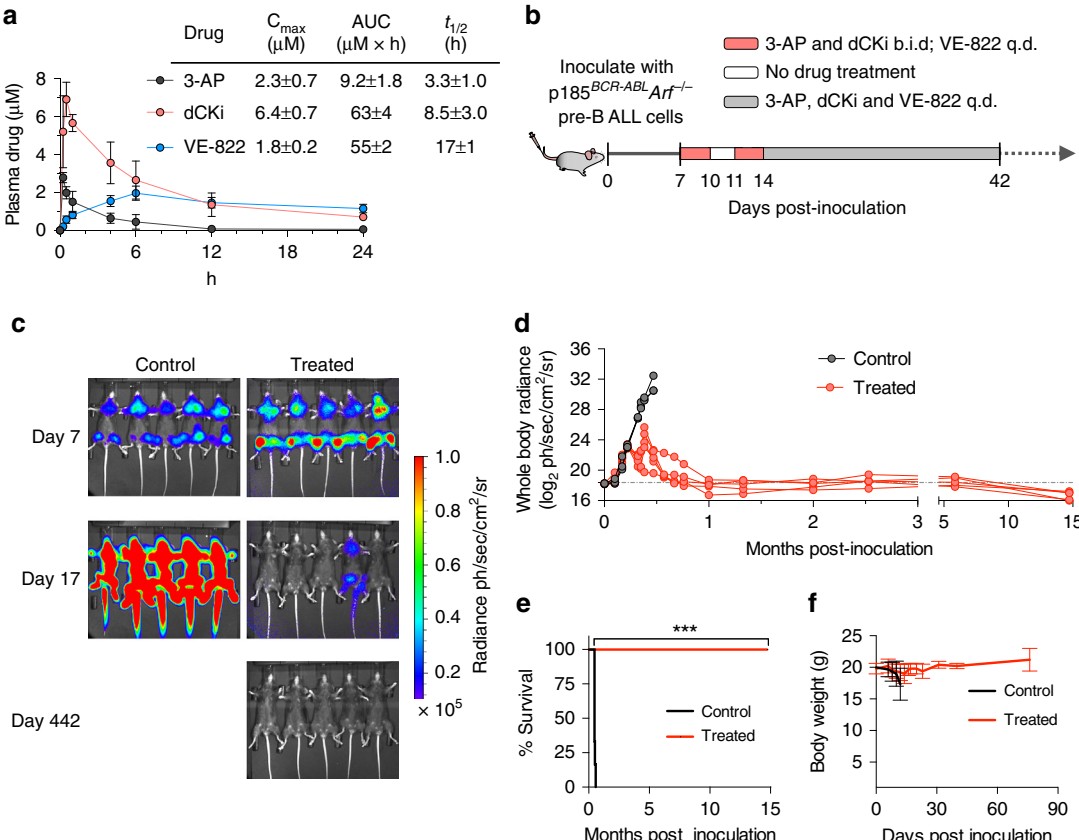

**Fig. 6** The triple combination therapy is effective and well-tolerated in a systemic primary B-ALL model. **a** Plasma pharmacokinetic parameters for 3-AP (15 mg kg$^{-1}$), VE-822 (40 mg kg$^{-1}$) and dCKi (50 mg kg$^{-1}$) in C57BL/6 mice ($n \geq 3$) after single dose oral co-administration (mean ± s.d., $n \geq 3$). **b** Doses and schedules for the triple combination therapy of leukemia bearing mice. **c, d** Bioluminescence images **c** and quantification of whole body radiance **d** of leukemia bearing mice treated with the combination therapy (treated, $n = 5$) or vehicle (control, $n = 5$) at indicated days after tumor inoculation. See also Supplementary Figs 11–13. **e, f** Kaplan–Meier survival analysis **e** and body weight measurements **f** of leukemia bearing mice treated with the combination therapy (treated, $n = 5$) or vehicle (control, $n = 5$). Median survival for the control group was 14 days after treatment initiation, whereas median survival for the treated group remains undefined (Mantel–Cox test). *$P < 0.05$; **$P < 0.01$; ***$P < 0.001$; ****$P < 0.0001$. q.d. once/day; b.i.d. twice/day

replication stress, and phosphorylated histone H2A.X on serine 139 (pH2A.X) as an indicator of DNA damage (Fig. 4b). RNR inhibition by 3-AP increased the percentage of ssDNA$^+$ cells by more than two-fold as early as 0.5 h after treatment, and by greater than three-fold at the 4 h time point. Combined inhibition of ATR and dCK also increased the ssDNA$^+$ and ssDNA$^+$;pH2A.X$^+$ cell populations at the 0.5 and 4 h time points. Addition of an RNR inhibitor resulted in a rapid and massive expansion of ssDNA$^+$ population at the 0.5 h time point compared to untreated cells, an effect which was further amplified at the later time point (Fig. 4b). Along with these changes, RNR inhibition triggered rapid induction of CHEK1 pS345, a direct downstream target of ATR (Fig. 4c). This PTM was abrogated by VE-822 treatment at the 0.5 h time point and partially inhibited at the 4 h time point. The rebound in CHEK1 pS345 expression at the 4 h time point was likely mediated by the ATR related kinase, ataxia telangiectasia mutated (ATM), which also phosphorylates CHEK1 on serine 345[53]. ATM was activated in cells treated with the triple combination therapy, as indicated by an increase in phosphorylation of CHEK2 on threonine 68 (pT68, Fig. 4c); a direct target of activated ATM. The induction of CHEK2 pT68 in cells treated with the triple combination therapy coincided with an increase in the pH2A.X$^+$ population (Fig. 4b) and with cleavage of the apoptotic markers caspase 8 (but not caspase 9), caspase 3, and Poly (ADP-ribose) polymerase (PARP) (Fig. 4c). Consistent with these observations, co-targeting ATR, dCK, and

RNR resulted in the highest percentage of apoptotic cells, as measured by Annexin V staining (Supplementary Fig. 8a). The cytotoxic effect of ATR inhibitors has been attributed to the induction of premature mitotic entry of cells undergoing DNA replication, an event that exacerbates the level of replication stress and DNA damage[54, 55]. Consistent with this model, ATR inhibition increased the percentage of S-phase CEM cells with phosphorylation of histone 3 on serine 10 (H3 pS10), a marker for mitotic kinase activation. This effect was significantly amplified in the triple combination therapy (Supplementary Fig. 8b).

**ATR inhibition alone is marginally effective in vivo**. To investigate the in vivo efficacy and tolerability of co-targeting alternative nucleotide biosynthetic pathways and ATR, we used a previously described primary *BCR-ABL*-expressing *Arf*-null pre-B (p185$^{BCR-ABL}$*Arf*$^{-/-}$) model which is difficult-to-treat and thought to be representative of the human disease[31, 33]. When compared with 31 cancer cell lines of different origins, p185$^{BCR-ABL}$*Arf*$^{-/-}$ cells were amongst the most sensitive to ATR inhibition by VE-822, with an IC$_{50}$ value of ~300 nM (Fig. 5a). However, despite its high potency in culture against pre-B-ALL cells, VE-822 alone was only marginally efficacious in vivo. C57BL/6 mice inoculated with luciferase expressing p185$^{BCR-ABL}$*Arf*$^{-/-}$ cells succumbed to disease within 17 days; all VE-822 treated

mice died of leukemia within 38 days after inoculation (Fig. 5b–d). We then investigated whether targeting the activities of de novo and salvage pathways can improve the efficacy of ATR inhibition in p185$^{BCR-ABL}$Arf$^{-/-}$ cells. Similar to the findings in the human T-ALL cells, targeting these biosynthetic pathways along with ATR was necessary to achieve maximal induction of cell death (Fig. 5e) and complete inhibition of cell growth (Supplementary Fig. 9a, b).

**Triple combination eradicates B-ALL in vivo.** To translate the above cell culture findings into an in vivo setting, we developed a new drug formulation consisting of PEG-200, Transcutol, Labrasol and Tween-80 blended in a ratio of 5:3:1:1 to solubilize three different drugs (3-AP, VE-822 and dCKi) and achieve therapeutically relevant plasma concentrations via oral delivery (Fig. 6a). Based on plasma pharmacokinetic parameters, 3-AP and dCKi were administered twice/day while VE-822 was administered once/day (Fig. 6b). Treatment was initiated on day 7 post inoculation of pre-B-ALL when all mice showed evidence of systemic disease, as indicated by whole body bioluminescence imaging (BLI, Fig. 6c, *top row, right panel*). While mice in the control group succumbed to disease within 17 days, mice in the combination treatment group had significantly lower disease burden on day 17 (Fig. 6c, d). All treated mice remained disease-free for 442 days after treatment withdrawal on day 42 (Fig. 6e). The combination therapy was well-tolerated, as indicated by maintenance of body weight during treatment (Fig. 6f) and long term survival (over 1 year and currently ongoing) without any detectable pathology. We also assessed the efficacy of the combination therapy when all three components were administered once daily. Although this therapeutic scheme appeared to be slightly less efficacious than the twice/day schedule for 3-AP and dCKi, it was well tolerated and four out of five mice had no detectable disease 313 days after treatment withdrawal (Supplementary Fig. 10). Importantly, removing the dCKi from the combination therapy significantly reduced the therapeutic efficacy in vivo (Supplementary Fig. 11), a result consistent with cell culture findings (Figs 4 and 5e, Supplementary Figs 8 and 9).

While BCR-ABL tyrosine kinase inhibitors are becoming standard care for patients with Philadelphia chromosome positive ALL[56], therapeutic resistance in pre-B-ALL is common and is caused by the rapid emergence of the T315I BCR-ABL kinase domain (gatekeeper) mutation[32, 56]. To test the combination of VE-822, dCKi and 3-AP against kinase inhibitor resistant ALL we generated p185$^{BCR-ABL}$Arf$^{-/-}$ T315I mutant cells by exposing leukemia bearing mice to dasatinib and harvesting drug-resistant cells from bone marrow (Supplementary Fig. 12a–c). Mice were inoculated with the T315I-positive cells and treated with the combination therapy (Supplementary Fig. 12d–g). The combination therapy was effective against the highly aggressive dasatinib resistant in vivo pre-B-ALL model with 13 out of 20 mice being disease-free over 365 days post inoculation of leukemia cells (Supplementary Fig. 12). To determine whether mice that did not achieve complete remissions harbor ALL cells that have acquired resistance to the triple combination, we harvested leukemia cells from the bone marrow of the moribund mice. These cells were then used to test the efficacy of the combination treatment in cell culture, and compare it with the original dasatinib-resistant p185$^{BCR-ABL}$Arf$^{-/-}$ pre-B-ALL cells. The harvested leukemia cells responded to the combination treatment as well as did the parental cells (Supplementary Fig. 13). One potential reason for the incomplete response in some of the treated mice is the rapid engraftment of p185$^{BCR-ABL}$Arf$^{-/-}$ T315I + pre-B-ALL cells in the brain coupled with the suboptimal penetrance of 3-AP and potentially, dCKi across the blood–brain-barrier.

**Discussion**

Here we show that ATR inhibition in leukemia cells reduces the output of both de novo and salvage pathways. However, significant remaining activities of both pathways were sufficient to prevent ATR inhibition-induced DNA replication shutdown in cell culture, and permit disease-induced lethality in a systemic mouse model of pre-B-ALL. Combining ATR inhibition with inhibitors of de novo (RNR) and salvage (dCK) rate-limiting enzymes led to rapid accumulation of ssDNA, a hallmark of replication stress[22], followed by extensive DNA damage, caspase-8 and PARP cleavage, and apoptosis. This synthetically lethal combination therapy was well-tolerated in vivo and promoted long-term disease-free survival in mice with systemic p185$^{BCR-ABL}$Arf$^{-/-}$ pre-B-ALL, as well as in a mouse model of targeted-therapy (dasatinib) pre-B-ALL resistance. Collectively our results quantify the control exerted by ATR on convergent nucleotide biosynthetic routes, and provide the rationale to co-target both signaling (ATR) and metabolic (RNR and dCK) mechanisms in acute leukemia for optimal therapeutic efficacy.

In mammalian cells dCTP and other dNTPs are present in low concentrations (1–50 pmol of dCTP/10$^6$ cells); amounts far below those required to complete one round of genome duplication (~ 1089 pmol of dCTP/10$^6$ cells)[14, 57]. This apparent discrepancy between dNTP supply and demand can be explained by a model in which dNTP production is tightly coupled with utilization for DNA synthesis—an 'on demand' model. According to this model, even small disruptions of dNTP production could significantly impact DNA integrity, unless the demand for dNTPs is reduced by preventing new origin firing, a process regulated by ATR and its effector kinases[43]. This prediction is supported by the synthetic lethality observed with ATR inhibition and pharmacological targeting of the *de novo* and salvage pathways observed in cell culture (Fig. 5e and Supplementary Figs 8, 9 and 13) and in vivo (Fig. 6c–f, Supplementary Figs 10–12).

To what extent are our findings of differential utilization of de novo and salvage pathways in leukemia applicable to solid tumors? An examination of three patient-derived primary cells point to significant nucleotide biosynthetic diversity in solid tumors (Supplementary Fig. 14a). HK-374 glioblastoma multiforme cells[58] preferentially use the de novo pathways to generate most of their dATP, dGTP and dCTP pools, with only the dTTP pool displaying a significant salvage component. A distinct nucleotide biosynthetic profile is present in two melanoma patient-derived primary cells, M299 and M417 (Supplementary Fig. 14a). In M299 cells, the salvage contribution to all four dNTPs exceeds 50%. Except for dATP, a similarly increased reliance on salvage biosynthesis was evident in M417 cells (Supplementary Fig. 14a). In contrast, de novo pyrimidine biosynthesis is almost completely absent in M417 cells, with greater than 98% of the dTTP and dCTP pools originating from salvage pathways. Stable isotope labeling studies using [$^{15}$N$_2$]orotate show that both M417 and M229 patient-derived primary cells efficiently used this substrate to produce dCTP, thereby suggesting a block in de novo pyrimidine biosynthesis in M417 cells either at the level of carbamoylphosphate synthetase II (CAD) or at the level of dihydroorotate dehydrogenase (Supplementary Fig. 14b, c). Indeed, M417 cells were subsequently found to harbor a homozygous nonsense mutation (Q140*) in CAD (Supplementary Fig. 14d). Predictably, the defect in de novo pyrimidine biosynthesis rendered M417 cells hypersensitive to dCK inhibition (Supplementary Fig. 14e). Confirmation of these findings in larger panels of cell lines and patient-derived primary samples would provide the rationale for developing new metabolically targeted therapies and clinically applicable biomarkers to define specific nucleotide biosynthetic subtypes

(e.g. predominant de novo, predominant salvage, and both de novo and salvage) in solid tumors. Such biomarkers could be provided by positron emission tomography imaging using new probes for nucleotide metabolism developed by us[59, 60] as well as by others[61].

PTMs can have profound and dynamic effects on the activity of metabolic networks[62]. Both dCK[30] and RRM2[44, 63] contain multiple PTMs. Among the four phosphorylation sites reported for dCK[30], phosphorylation on serine 74 (pS74) is an important determinant of dCK substrate specificity and catalytic activity[29]. Both ATR and ATM have been reported to phosphorylate dCK on S74 in response to replication stress and DNA damage respectively[28, 64]. Our data show that the ATR inhibitor VE-822 reduces dCK activity by ~ 33%, an effect likely attributed to a two-fold decrease in dCK pS74 (Fig. 2f). However, it remains unknown what kinases are responsible for the remaining dCK pS74 present in VE-822-treated leukemia cells. ATM is a potential candidate, given that phosphorylation of CHEK2 on T68 is detected in ATR inhibited cells (Fig. 4c). Further studies are needed to confirm this hypothesis. In this context, it may also be informative to analyze the activity of dCK following ATR inhibition in tumors harboring inactivating mutations in ATM[65–67]. Concerning RNR regulation, the observed consequences of ATR inhibition in T-ALL cells include the following: a decrease in RRM2 protein levels (Fig. 2c, d), reduced RRM1 protein levels (Fig. 2c, d), and a significant reduction in RRM2 pT33 (Fig. 2e), a CDK-mediated phosphorylation event which promotes RRM2 proteasomal degradation via interactions with the SCF$^{Cyclin\ F}$ ubiquitin ligase complex[44]. It is currently unknown whether the reduction in RRM2 levels by ATR inhibition in leukemia cells occurs via a post-translational mechanism concerning protein stability and/or by a transcriptional mechanism downstream of E2F family members[68, 69]. Moreover, further studies are needed to determine the significance of the observed reductions in RRM1 protein levels following ATR inhibition (Fig. 2c, d) and to identify the mechanism(s) responsible for this effect.

Our results provide quantitative insights into alterations of nucleotide biosynthetic pathways induced in leukemia cells by inhibiting ATR and rate-limiting de novo (RNR) and salvage (dCK) enzymes. These findings support a new therapeutic strategy that uses existing inhibitors to exploit the dependency of leukemia cells on intact de novo and salvage biosynthetic pathways and replication stress response mechanisms. Further refinements in this strategy and expanding its applicability beyond leukemia may come from follow-up studies to define clinically applicable companion biomarkers capable of delineating nucleotide biosynthetic and replication stress subtypes that are predictive of responses in human tumors.

## Methods

**Cell culture and culture conditions.** Leukemia cell lines: CCRF-CEM, EL4, Jurkat, Molt-4, CEM-R, THP-1, HL-60, TF-1, MV-4-11, HH, HuT 78, K-562; ovarian cancer cell lines: Hey-T30, PA-1, Caov-3, OVCAR-5, IGROV-1, A2780; and pancreatic adenocarcinoma cell lines: BxPC-3, MIA PaCa-2, Hs 766 T, AsPC-1 and PANC-1 were obtained from American Type Culture Collection (ATCC). Various cell lines were kind gifts: Nalm-6 (Michael Teitell, UCLA), p185$^{BCR-ABL}$Arf$^{-/-}$ pre-B cells (N. Boulos, CERN Foundation), L3.6pl (David Dawson, UCLA), KPC (Guido Eibl, UCLA), and patient-derived primary cells of leukemia COG332 (Yong-Mi Kim, USC). All hepatocellular carcinoma cell lines were gifts from Ali Zarrinpar (Univ. of Florida): SNU-475, PRC/PRF/5, SNU-449, Hep 3B, Hep G2, and SK-HEP-1. Patient-derived primary cells of glioblastoma (HK-374) and melanoma (M299 and M417) were derived in the labs of Drs. Kornblum (UCLA) and Ribas, respectively. With a few exceptions, cell lines were cultured in RPMI-1640 (Corning) containing 10% fetal bovine serum (FBS, Omega Scientific) and were grown at 37 °C, 20% O$_2$ and 5% CO$_2$. p185$^{BCR-ABL}$Arf$^{-/-}$ pre-B cells were cultured in RPMI-1640 containing 10% FBS and 0.1% β-mercaptoethanol. HK374 was cultured in DMEM-F12 (Invitrogen) containing B27 supplement (Life Technologies), 20 ng ml$^{-1}$ basic fibroblast growth factor (bFGF; Peprotech),

50 ng ml$^{-1}$ epidermal growth factor (EGF; Life Technologies), penicillin/streptomycin (Invitrogen), Glutamax (Invitrogen) and 5 µg ml$^{-1}$ heparin (Sigma-Aldrich). All cultured cells, except HK374, were incubated in antibiotic free media and were regularly tested for mycoplasma contamination using MycoAlert kit (Lonza) following the manufacturer's instructions, except that the reagents were diluted 1:4 from their recommended amount.

**Proliferation assays.** Cells were plated in 384-well plates (1000 cells/well for suspension cell lines and 500 cells/well for adherent cell lines in 30 µl volume). Drugs were serially diluted to the desired concentrations and an equivalent volume of DMSO was added to vehicle control. Ten microliter of the 4× diluted drugs were added to each well. Following 72 h incubation, ATP content was measured using CellTiter-Glo reagent according to manufacturer's instructions (Promega, CellTiter-Glo Luminescent Cell Viability Assay), and analyzed by SpectraMax luminometer (Molecular Devices). IC$_{50}$ values, concentrations required to inhibit proliferation by 50% compared to DMSO treated cells, were calculated using Prism 6.0 h (Graphpad Software).

**Isotopic labeling in cell culture.** Cells were transferred into RPMI-1640 without glucose and supplemented with 10% dialyzed FBS (Gibco) containing the following labeled substrates: precursors for de novo [U-$^{13}$C$_6$]glucose (Sigma-Aldrich, 389374) at 11 mM; precursors for purine salvage [U-$^{13}$C$_{10}$,$^{15}$N$_5$]dA (Cambridge Isotopes, CIL 3896), [$^{15}$N$_5$]dA (Cambridge Isotopes, NLM-3895) and [$^{15}$N$_5$]dG (Cambridge Isotopes, NLM-3899) at 5 µM or as indicated; and precursors for pyrimidine salvage: [U-$^{13}$C$_9$,$^{15}$N$_3$]dC (Silantes, 124603602) and [U-$^{13}$C$_{10}$,$^{15}$N$_2$]dT (Cambridge Isotopes, CNLM-3902) at 5 µM and [$^{15}$N$_2$]orotate (Cambridge Isotopes, NLM-1048) at indicated concentrations. The cells were incubated for 12 h or as indicated before sample collection and processing.

**Western blot.** Cells were lysed using RIPA buffer supplemented with protease (ThermoFisher, 78,430) and phosphatase (ThermoFisher, 78,420) inhibitors, scraped, sonicated, and centrifuged (20,000 × g at 4 °C). Protein concentrations in the supernatant were determined using the Micro BCA Protein Assay kit (Thermo), and equal amounts of protein were resolved on pre-made Bis-Tris polyacrylamide gels (Life Technologies). Primary antibodies: pS345 CHEK1 (Cell signaling, #2348 L, 1:1000), pT68 CHEK2 (Cell signaling, #2197 S, 1:1000), pS139 H2A.X (Millipore, 05-636, 1:1000), clvd. Casp8 (Cell signaling, #8592, 1:1000), clvd. Casp9 (Cell signaling, #9502, 1:1000), clvd. Casp3 (Cell signaling, #9662, 1:1000), clvd. PARP (Cell signaling, #5625 P, 1:1000), and anti-actin (Cell Signaling Technology, 9470, 1:10,000). Primary antibodies were stored in 5% BSA (Sigma-Aldrich) and 0.1% NaN$_3$ in TBST solution. Anti-rabbit IgG HRP-linked (Cell Signaling Technology, 7074, 1:2500) and anti-mouse IgG HRP-linked (Cell Signaling Technology, 7076, 1:2500) were used as secondary antibodies. Chemiluminescent substrates (ThermoFisher Scientific, 34,077 and 34,095) and autoradiography film (Denville) were used for detection.

**Drugs.** The following drugs were used: Pablociclib (Selleckchem, S1116, 1 µM), VE-822 (ApeXBio, B1381, 1 µM or as indicated), DI-82 (dCKi$^2$, 1 µM), thymidine (Sigma-Aldrich, T1895, as indicated), hydroxyurea (Sigma-Aldrich, H8627, as indicated), gallium maltolate (Nanoman Industries, CN-GAM-02-1G00-A00, as indicated), 3-AP (ApeXBio, custom, 500 nM or as indicated), Pentostatin (Santa Cruz Biotechnology, sc-204177, 10 µM), forodesine or BCX-1777 (Chemscene, CS-3781, 100 nM), and dasatinib (LC Laboratories, D-3307, 1 nM).

**Cell cycle kinetics.** CEM T-ALL cells were synchronized in G1 phase, and then treated as indicated. Cells were pulsed with EdU 1 h before collection at different time points. Cells were fixed 4% paraformaldehyde, permeabilized with perm/wash reagent (Invitrogen), stained with Azide-AF647 (using click-chemistry, Invitrogen; Click-iT EdU Flow cytometry kit, #C10634) and FxCycle-Violet (Invitrogen), and then analyzed flow cytometry (a detailed description is available in the Supplementary Information).

**Metabolite profiling.** See Supplementary Information for details on sample preparation, data collection and analysis.

**Global proteomic and phosphoproteomic analyses.** See Supplementary Information for details on sample preparation, data collection and analysis.

**ssDNA and pH2A.X Measurements.** Treated CEM T-ALL cells were fixed with ice-cold methanol:PBS (6:1 v/v). Staining with F7-26 ssDNA monoclonal antibody (mouse) was performed according to manufacturer's instructions (EMD Millipore, #MAB3299) followed by secondary antibody IgM-PE. Subsequently, cells were stained with the phospho-Histone H2A.X (pS139 H2A.X) specific antibody conjugated to FITC (EMD Millipore, #05-636) and DAPI, and then analyzed by flow cytometry (a detailed description is available in the Supplementary Information).

**Western blots**. See Supplementary Information for details on sample preparation and antibody information. Images have been cropped for presentation. Full-size images are presented in Supplementary Fig. 15.

**Apoptosis assay**. Apoptosis and cell death were assayed using Annexin V-FITC and PI according to manufacturer's instructions (FITC Annexin V Apoptosis Detection Kit, BD Sciences, #556570).

**In vivo leukemia models and treatment regimes**. All animal studies were approved by the UCLA Animal research committee (ARC). For development of systemic leukemia model, C57Bl/6 female mice were injected intravenously with 50,000 firefly luciferase expressing p185$^{BCR-ABL}$Arf$^{-/-}$ pre-B-ALL cells[12, 31]. The leukemic burden was monitored using bioluminescence imaging. All drug treatments were performed in an oral formulation consisting of PEG-200, Transcutol, Labrasol and Tween-80 blended in a ratio of 5:3:1:1. The oral gavage volume was 50 μl per dose (a detailed description is available in the Supplementary Information).

**Pharmacokinetic measurements**. 3-AP, dCKi, VE-822 plasma concentrations were assessed at 0.5, 1, 2, 4, 6, 12 and 24 h following oral administration. Blood samples were collected in heparin-EDTA tubes by the retro-orbital technique and spun down to collect the supernatants. Four parts of methanol were added to plasma samples to precipitate proteins, and the supernatant was collected. Twenty microliter samples were injected onto a reverse phase column equilibrated in water 0.1% formic acid for LC-MS/MS-MRM analysis in positive ion mode. Plasma concentrations were determined by comparison to previously generated standard curves. A population pharmacokinetic modeling (NONMEM v. 7.2) with first-order conditional estimation (FOCE) was used to characterize the C$_{max}$, half-life, and area under the curve (AUC) for 3-AP, dCKi, and VE-822. 3-AP and VE-822 were best described with a one compartment model while dCKi was best described by a two-compartment model. All three drugs were well described with first order absorption and a proportional error model. Clearance and volume were scaled by weight.

**FACS analyses**. All flow cytometry data were acquired on a five-laser LSRII cytometer (BD), and analyzed using the FlowJo software (Tree Star). See Supplementary Information for details on propidium iodide, cell cycle by EdU, pH2A.X, ssDNA, phospho-histone 3 and Annexin V staining.

**Animal studies**. Mice were housed under specific pathogen-free conditions and were treated in accordance with UCLA Animal Research Committee protocol guidelines. All C57BL/6 female mice were purchased from the UCLA Radiation Oncology breeding colony. VE-822 (ApeXBio, 40 mg kg$^{-1}$), 3-AP (ApeXBio, dosage as indicated) and DI-82 (dCKi, Sundia Pharmaceuticals, 50 mg kg$^{-1}$) were administered by intraperitoneal (i.p.) injections or oral gavage to recipient animals. For oral administration, single agent or combination of drugs were solubilized in a formulation consisting of the following: PEG-200: Transcutol: Labrasol: Tween-80 mixed in 5:3:1:1 ratio. For i.p. administration, the drugs were solubilized in PEG-400 and 1 mM Tris-HCl in a 1:1 (v:v) ratio. Dasatinib (LC Laboratories) was solubilized in 80 mM citric acid (pH 3.1) and administered at a dose of 10 mg kg$^{-1}$ by oral gavage. $2 \times 10^5$ luciferase expressing p185$^{BCR-ABL}$Arf$^{-/-}$ pre-B-ALL cells were injected intravenously into C57BL/6 female mice for leukemia induction. The treatment was started 6 or 7 days after the intravenous inoculation of leukemia initiating cells, when animals had developed a significant leukemic burden as monitored by bioluminescence imaging (IVIS Bioluminescence Imaging scanner). The dosing schedules are indicated in the text and figure legends. The mice were observed daily and those that became moribund during the trials, (paralysis of hind limbs, significant body weight loss) were sacrificed immediately. Kaplan–Meier curves and bioluminescence quantifications were generated using Prism 6.0 h (Graphpad Software).

**Bioluminescence imaging (BLI)**. Mice were anesthetized with 2% isoflurane followed by intraperitoneal injection of 50 μl (50 mg ml$^{-1}$) substrate D-luciferin (Sigma, #L9504). The mice were imaged with the IVIS 100 Bioluminescence Imaging scanner 10 min after luciferin administration. All mice were imaged in groups of five with 1-minute exposure time, and the images were acquired at low binning.

**Statistical analyses**. Data are presented as means ± s.d. with indicated biological replicates. Comparisons of two groups were calculated using indicated unpaired or paired two-tailed Student's *t*-test and *P* values less than 0.05 were considered significant. For some experiments, generated mean normalized values (ratios from two groups, treated to untreated) were compared to the hypothetical value 1 (indicating equal values between treated and untreated), calculated using one-sample *t*-test, and *P* values less than 0.05 were considered significant. Comparisons of more than two groups were calculated using one-way ANOVA followed by Bonferroni's multiple comparison tests, and *P* values less than 0.05 /*m*, where *m* is the total number of possible comparisons, were considered significant.

All statistical analysis and generated graphs were performed either in R or Graphpad Prism 6.0 h.

The rates of dCTP incorporation into the DNA were determined using linear regression with procedure GLM in SAS (v. 9.4). DNA % label was transformed using a shifted log addition of 1 due to the presence of zeros in the data set. Rates were calculated using the slope of the regression line for the log transformed data and are presented in the table with the associated standard errors, along with the ratio between the rate of incorporation into DNA by the de novo and salvage pathways (Table 1).

**Data availability**. The proteomic and phosphoproteomic data have been deposited in the Proteome Xchange database under accession code PXD006702. The authors declare that all other data supporting the findings of this study are available within the article and its Supplementary Information Files or from the corresponding author upon request.

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

## Acknowledgements

We acknowledge the following people for their critical review of the manuscript: Irmina Gawlas and Andreea Stuparu (UCLA), Nathan Mata and Neil Bajpayee (Trethera Corporation) and Ting-Ting Wu (UCLA). We acknowledge the excellent technical assistance provided by Xuemeng Wang, Jimmy Bazzy, Anthony E. Cabebe, Thotsophon Taechariyakul, Taylor Hulahan, and Emily Hua. This work was funded by National Cancer Institute grant R01 CA187678 (CGR), the US Department of Energy Office of Science award DE-SC0012353 (JC and CGR), Jonsson Comprehensive Cancer Center Foundation/UCLA Impact grant (CGR), and Keck Grant (CGR, JC, and TD). JW acknowledges support from the UCSD/UCLA NIDDK Diabetes Research Center P30 DK063491. TML was supported by the UCLA Scholars in Oncologic Molecular Imaging program (SOMI), under National Cancer Institute award R25 CA098010.

## Author contributions

T.M.L., S.P., J.R.C., T.R.D., JC, H.R.H and C.G.R., were involved in study conception and experimental design, data analysis, and writing the manuscript with editorial assistance from the other authors. S.P., E.R.A., W.K., N.T.U. and C.M.C. performed experiments. T.M.L., J.R.C., N.T.U., C.M.C., D.M., J.W., K.F.F. and D.B. designed and performed the mass spectrometry experiments and analyzed the data. S.P., L.W., E.R.A. and M.N. performed mouse treatment studies and PK analyses. P.R. and S.P. developed the drug formulation for oral delivery. J.Z., A.R. and H.I.K provided key reagents and assisted with data analysis.

## Additional information

**Competing interests:** The authors declare the following competing financial interest(s): C.G.R. and J.C. are co-founders of Trethera Corporation. They and the University of California hold equity in Trethera Corporation. The University of California has patented additional intellectual property for small molecule dCK inhibitors invented by C.G.R., J.C., S.P. and T.M.L. This intellectual property has been licensed by Trethera Corporation.

