## [Peer Review File · Nature Communications]

Reviewers' comments:

Reviewer #1 (Remarks to the Author):

Le, Poddar, Capri et al.

In this manuscript, the authors examine the role of ATR in regulating the de novo and salvage nucleotide biosynthetic pathways through regulation of their rate-limiting enzymes and show how this relationship can be utilized as a therapy for leukemia. The authors begin by demonstrating that 1) dual inhibition of ATR and dCK slows S phase entry, 2) ATR inhibition reduces de novo dCTP synthesis mediated by glucose utilization, and 3) ATR inhibition reduces utilization of dCTP generated by both biosynthesis pathways for DNA replication. The effect of ATR inhibition on the generation of dCTP from these pathways is quantified to high precision. From these and other data, the authors argue that ATR regulates enzymes in both biosynthesis pathways. Additionally, the authors determine which RNR inhibitor is most potent and study their effects on dCTP biosynthesis pathways. Furthermore, they show that the salvage pathway compensates for a decrease in the de novo pathway. These findings could inform patient treatment regimens as such compensation could be a means of resistance in the clinic. Using a combination of ATR, dCK and RNR inhibitors, the authors show a synergistic effect on DNA replication, DSB formation and ultimately apoptosis using the triple combination therapy. Finally, the authors assess the clinical relevance of their finding in a B-ALL mouse model, in which they conclude that this triple combination therapy would be beneficial for B-ALL patients, including for patients who develop resistance to BCR-ABL kinase inhibitors.

The findings described in this paper are novel and of great interest to the field. Although a relationship between nucleotide production pathways and ATR has been established, it has yet to be defined in the context of leukemia. In addition, the therapeutic potential of this combination could have an impact in the clinic due to the large therapeutic window. Generally, the studies are performed rigorously; however, some of the claims need to be further substantiated with a few additional experiments as suggested below (Major Points). Other issues to be addressed are also listed (Minor Points).

Major Points:

1. It is possible that some of the effects of ATR inhibition on the rate of nucleotide (CTP) incorporation into DNA could be due to replication fork collapse and delays in restart, which have been shown to occur in ATR inhibited and suppressed cells. The authors should address this possibility in the text.
2. For Figure 1d/e, it would be helpful to use a second drug or to use conditional gene suppression approach (dox-inducible shRNA, TRIPZ) to show that these effects are not due to off target effects. However, it is probably on target due to low drug concentrations used (1 μ M).
3. For Figure 1d/e, it would be interesting to show that the delay in entry is due to a decrease in nucleotide biosynthesis by rescuing it with nucleotide addback. This would help show that the effects are not due to an off target of the inhibitors.
4. For Figure 3c, the authors should show that knocking down RRM1/2 to levels similar to that seen after ATR inhibition leads to the same effects on dCTP biosynthesis (i.e. these levels of reduction have a biological effect).
5. For Figure 3, the authors claim that ATR inhibition leads to differential utilization of dCTP generated by biosynthesis pathways as a result of its effects on enzymes that regulate these pathways. The evidence presented is correlative and needs to be better tested. Follow up studies testing that ATR inhibition directly leads to decreases in pS74 dCK and pT33 RRM2 and

establishing the contribution of these phosphosite modifications on dCMP biosynthesis and total RRM2 and dCK levels are necessary to show causation.

6. It would be interesting to compare the levels of the biosynthesis pathways in the cells of the dasatinib resistant pre-B ALL in the mice responding to treatment (13 of 20) vs the ones that don't respond (7 of 20). This experiment would help determine if there are other pathways at play or if there is another resistance mechanism that compensates for the biosynthesis pathway knockdown. This approach would provide important information for selecting patients who would gain a clinical benefit from this treatment (i.e. clinical biomarkers to determine which patients would benefit from this therapy).

Minor Points:

1. Figure 2a, typo of "unlabeled" label above the relative abundance graph.
2. Figure 3g, align "drug, level, p sites..." labels in table. Also add line under RNR, dCK and substrate utilization labels to make it more clear that "level" and phosphosites belong to those labels above.
3. Figure 5b, H2AFX is more commonly referred to as H2AX so if you wanted more room on your y axis in the graphs/due to more common nomenclature, you could change that to H2AX. Especially, because you use a different nomenclature in Supp Fig 3g (H2A.X). Better to pick one and stay consistent.
4. Figure 6d, "drug holiday" could be replaced with "no drug treatment" or something to that effect to increase clarity. Same comment applies to sup fig 7c.
5. Supp Fig 3g, keep protein name and phosphosite designation consistent throughout (ie use "pS139 H2AX" and use "CHEK1").
6. Supp Fig 3 and 4 legends include a legend for a panel h which does not exist. I believe you forgot to label your DNA-A/DNA-G bar graphs as "d" and as a result all your panels thereafter have the wrong letter designation.
7. Supp Fig 4c, in figure legend "(c)" is not bolded.
8. Supp Fig 5, add dose response curves for each drug concentration and their corresponding drug dosage.
9. Supp Fig 6, add corresponding flow cytometry plots for Annexin V staining that are summarized in the bar graph of Supp Fig 6a.
10. Supp Fig 6c, please show all of the flow graphs: include the the single drug treatments and the dual drug treatments. Particularly those of ATR inhibition since you highlight these in the text as a source of premature entry of cells into mitosis.
11. Figure legend for Supp Fig 6 is inaccurate (b and c descriptions are flipped).
12. Supp Fig 8, in the legend, the letters are offset again. Add "(c)" before "measurement of apoptosis..." in legend to fix and adjust all letters thereafter.
13. In the last part of the result section, when discussing gatekeeper mutation, you refer to Fig 8a,b but you mean supplemental Fig 8.

Reviewer #2 (Remarks to the Author):

In this study, Le and colleagues claim that ATR inhibition decreases the output of both de novo and salvage nucleotide pathways. The authors combined ATR inhibitors with inhibitors of RNR and dCK which lead to single stranded DNA, DNA damage and consequently apoptosis. Certainly, these two metabolic enzymes are required for DNA synthesis. However, the authors categorized RNR as a metabolic enzyme belonging to de novo nucleotide synthesis and dCK involved in the salvage pathway. While dCK is a metabolic enzyme of the salvage pathway, RNR belongs to both salvage and de novo. Indeed RNR produces deoxyribonucleotide from ribonucleotide, but this involved metabolic output from de novo and salvage pathways. The authors should address this issue. Although the study is interesting, the major issue of this study is the lack of molecular mechanism explaining how ATR inhibition alters the nucleotide synthesis pathway.

Points:

Fig. 1: The authors use ¹³C₆-glucose to assess the effect of ATR inhibition on de novo nucleotide synthesis. The major issue of this experiment is that glucose is not only a substrate for the de novo pathways, it is also a substrate for the salvage pathway. Indeed, glucose can be converted through the pentose phosphate pathway into PRPP which is a substrate for de novo purine and pyrimidine synthesis but also for the salvage pathway (ie, APRT/HPRT). Assessing the incorporation of 13-carbon from ¹³C-glucose into UTP, CTP and CDP does not give a read out of the activity of de novo nucleotide synthesis pathway. The authors should show other metabolites specific for the de novo pathway (ie, AICAR, Adenylosuccinate (de novo purine) and N-carbamoyl-aspartate, orotate, or orotidine monophosphate (de novo pyrimidine)). If it is not possible, the authors should perform the tracing experiment with another substrate specific for de novo nucleotide synthesis. Plus, the authors should measure glucose uptake.

Fig.2: Incorporation of ¹⁵N-¹³C from ¹⁵N-¹³C-deoxycytidine (deoxynucleoside) into dCTP is specific for the salvage pathway since it is bypassing the need for glucose, however, other method should be used to assess the de novo pathway.

Fig.3: The phosphorylation of RRM2 on T33 is interesting. The authors should demonstrate whether ATR directly phosphorylate that site. The author should demonstrate whether the regulation of RRM2 by ATR is transcriptional or post-translational or both. Utilization of genetic approaches to confirm this regulation is needed (siRNA anti ATR or siRNA anti-dCK or combination).

Reviewer #3 (Remarks to the Author):

The manuscript by Le et al provides a beautiful description of the contributions of de novo and salvage pathways for pyrimidines in leukemia cells, and the role of ATR in regulating these pathways. The experiments are very detailed and the combined proteomics and metabolomics approaches provide a very clear description of the activity in each of these pathways, and the impact of specific inhibitors on those pathways. The in vitro and in vivo activity studies demonstrate the promise of inhibiting these pathways for cancer treatment, however, as nicely described in the discussion section, a cell-type-specific approach will likely be needed to account for the metabolic heterogeneity in cancer cells.

The observations described in the manuscript appear robust. Some of the conclusions and interpretation might be slightly overstated as described below:

1. Whilst ATR inhibition causes statistically significant decreases in nucleotide synthesis and salvage, it is only a small decrease, and the remaining levels are far more than 'residual'.

2. It is not clear that the impact of ATR inhibition on cell survival is primarily due to nucleotide depletion. It is possible that the primary action of ATR inhibition is independent of nucleotide depletion, but that the combination of ATR inhibition and metabolic inhibition has an additive effect resulting in decreased cell survival. Some critical controls are not shown that would further support the claim of synthetic lethality. In particular for the in vitro tests (fig 5b-c) the combination of metabolic inhibitors (D + AP) is not included. For the in vivo study there is no comparison to single or dual inhibitor combinations to confirm that this triple-therapy is actually necessary.

3. The metabolomics (untargeted metabolite profiling) experiment is not described in the methods. How were these samples extracted and analysed? Whilst a volcano plot is shown, only the nucleotides are labelled, whereas there are several metabolites that exhibit a larger change that are not labelled. If those metabolites are not nucleotides then it would argue against the primary mechanism of ATR inhibition involving nucleotide disruption (unlike the proteomics data that clearly shows RRM2 and TYMS and the most perturbed proteins). Furthermore, only the isotope enriched metabolites are shown. It would be interesting to see the full metabolomics data of relative metabolite levels to gauge the importance of nucleotide perturbation in the overall impact of ATR inhibition.

4. For the targeted metabolite labelling experiments it is convenient and probably appropriate to simplify the analysis to the three major species representing the unlabelled, de novo and salvage pathways. However, one would expect a number of other isotopologues to exist, e.g. label incorporation into the uracil moiety in de novo synthesis, or recombination of unlabelled (d)ribose with labelled cytosine (if there is active phosphorylase activity) for the salvage pathway. Could you provide an overview of the complete labelling pattern derived from these species in the supplementary data, and whether that changes after incubation with inhibitors?

5. For the glucose labelling experiment it would be interesting to see the % label incorporation for the species that come between glucose and the nucleotides in the metabolic pathways. At a minimum PRPP or ribose 5-phosphate should be shown to confirm that any change in label incorporation is due to the de novo nucleotide synthesis pathway rather than isotope dilution in central carbon pathways.

6. For the proteomics experiments it is not clear how the dimethyl labelling experiments were designed. The manuscript describes triplex labelling, but there are 4 sample groups. It mentions $n=3$, but according to fig 3a you'd end up with $n=6$ for the NT and V groups. Please clarify which data is included (and reasons for excluding data if this occurred). Furthermore, for labelling-based quantification experiments it is critical to switch the labelling to avoid labelling artefacts. Please describe how this was done (or complete this extra experiment if required). Also, how was the False Discovery Rate calculated for the statistical analysis of the proteomics data (it is mentioned in the results, but not in the methods).

7. In the phosphoproteomics, the decreased level of pRRM2 is described, however, total levels of RRM2 were also down. Therefore, one would expect the pRRM2 levels to also decrease. Therefore this is not a PTM modification (i.e. the % of phosphorylated RRM2 is the same), just a decrease in total RRM2 abundance.

8. Some technical details are missing from the methods: (i) how many cells were used per sample for the metabolomics and proteomics studies (ii) details for the non-targeted LCMS method (as mentioned above) and (iii) Please justify the normalisation strategy whereby the TIC at a specified retention time was used... if there was matrix effect at a specific retention time this might result in enhanced TIC but signal suppression, in which case this normalisation strategy would actually worsen the quantitative accuracy.

Reviewer #1

1. It is possible that some of the effects of ATR inhibition on the rate of nucleotide (CTP) incorporation into DNA could be due to replication fork collapse and delays in restart, which have been shown to occur in ATR inhibited and suppressed cells. The authors should address this possibility in the text.

We agree that this may be possible and we have expanded the results section to include references that investigate in detail the impact of inhibition of ATR signaling on origin firing and replication fork restart. The text now reads as follows:

“ATR inhibition reduced the DNA incorporation of both *de novo* and salvage produced dCTP, yielding a combined 30% reduction in overall DNA labeling compared to untreated cells at the 12 h time point (Fig. 1g and Table 1). This reduction is consistent with data in the literature showing replication fork collapse and delays in restarting DNA replication following ATR inhibition in other cell types^{40–43}.”

Cited references:

40. Eykelenboom, J. K. et al. ATR activates the S-M checkpoint during unperturbed growth to ensure sufficient replication prior to mitotic onset. *Cell Rep* **5**, 1095-1107 (2013).
41. Koundrioukoff, S. et al. Stepwise activation of the ATR signaling pathway upon increasing replication stress impacts fragile site integrity. *PLoS Genet* **9**, e1003643 (2013).
42. Marheineke, K. & Hyrien, O. Control of replication origin density and firing time in *Xenopus* egg extracts: role of a caffeine-sensitive, ATR-dependent checkpoint. *J Biol Chem* **279**, 28071-28081 (2004).
43. Shechter, D., Costanzo, V. & Gautier, J. ATR and ATM regulate the timing of DNA replication origin firing. *Nat Cell Biol* **6**, 648-655 (2004).

2. For Figure 1d/e, it would be helpful to use a second drug or to use conditional gene suppression approach (dox-inducible shRNA, TRIPZ) to show that these effects are not due to off target effects. However, it is probably on target due to low drug concentrations used (1 μ M).

We thank the reviewer for raising this point. We have selected VE-822 over other ATR inhibitors because this compound has an excellent selectivity profile when screened against the human kinome, and >100-fold cellular selectivity for ATR compared to other members of the PIKK family (including ATM and DNA-PK, ref #35, Fokas et al., 2014).

3. For Figure 1d/e, it would be interesting to show that the delay in entry is due to a decrease in nucleotide biosynthesis by rescuing it with nucleotide addback. This would help show that the effects are not due to an off target of the inhibitors.

We agree with the reviewer that the nucleotide addback experiment would yield useful information about the mechanism by which ATR inhibition interferes with cell cycle kinetics. Therefore, we performed the experiments suggested by the reviewer in both synchronous and asynchronous leukemia cells treated with the ATR inhibitor. As shown in the new Supplementary Figure 2, addition of a balanced mixture of dNTPs to the culture media partially rescued the defects in cell cycle progression induced by ATR inhibition. The effects of exogenously added dNTPs were reversed by dCK inhibition, indicating that these nucleotides are likely converted to the corresponding nucleosides in the extracellular environment, presumably by ectonucleotidases. The resulting nucleosides are then transported across the plasma membrane and must be phosphorylated by nucleoside kinases such as dCK to exert their protective effects in ATR inhibited cells.

Moreover, we think that the inability of exogenously added dNTPs to completely rescue the alterations in cell cycle kinetics induced by ATR inhibition may reflect the fact that, in T-ALL cells, purine deoxyribonucleosides are rapidly catabolized to hypoxanthine (Hx) via the actions of adenosine deaminase (ADA) and purine nucleoside phosphorylase (PNP), as shown in Supplementary Figures 5 and 6. While Hx can be salvaged by hypoxanthine phosphoribosyltransferase (HPRT), conversion of the resulting ribonucleotides to the corresponding purine dNTPs requires RNR activity, which as shown by us and others is reduced following ATR inhibition. The mechanisms described in the first comment by Reviewer #1 (*replication fork collapse and delays*

in restart, which have been shown to occur in ATR inhibited and suppressed cells”) may also prevent a full rescue of cell cycle kinetics by nucleotide supplementation. We think these observations, resulting from the suggestion of the referee, add to the manuscript; we thank him/her for this suggestion.

4. *For Figure 3c, the authors should show that knocking down RRM1/2 to levels similar to that seen after ATR inhibition leads to the same effects on dCTP biosynthesis (i.e. these levels of reduction have a biological effect).*

This is also a good point raised by the reviewer. To address the point, we have generated RRM2 knockdown CEM cells (shRNA^{RRM2}). An analysis of these cells is included in new Supplementary Fig. 7. RRM2 levels in the CEM shRNA cells were reduced by 35–50%, as determined by quantitative nLC-MS/MS and intracellular flow cytometry analyses (Supplementary Fig. 7a and 7b). The CEM shRNA^{RRM2} cells exhibited approximately 30% lower incorporation of *de novo* synthesized dCTP into newly replicated DNA (Supplementary Fig. 7c), which is comparable with the effects of pharmacological inhibition of ATR (Fig. 1g). These findings suggest that the RRM2 regulation by ATR is an important determinant of *de novo* dCTP biosynthesis in T-ALL cells. Since ATR inhibition reduced RRM2 levels by ~20% at the 12 h time point, it is likely that there are other mechanisms by which this replication stress response kinase regulates *de novo* dCTP biosynthesis. These additional mechanisms could include reduced levels of the large RNR subunit, RRM1 (Fig. 2c and 2d), and/or changes in yet to be identified regulatory PTMs in RRM1 and RRM2. Again, we think that these observations, resulting from the suggestion of the referee, add to the manuscript; we thank him/her for this suggestion.

5. *For Figure 3, the authors claim that ATR inhibition leads to differential utilization of dCTP generated by biosynthesis pathways as a result of its effects on enzymes that regulate these pathways. The evidence presented is correlative and needs to be better tested. Follow up studies testing that ATR inhibition directly leads to decreases in pS74 dCK and pT33 RRM2 and establishing the contribution of these phosphosite modifications on dCMP biosynthesis and total RRM2 and dCK levels are necessary to show causation.*

We agree that our observations are correlative, and do not prove causality. The penultimate paragraph in the Discussion section of the original manuscript acknowledged that the observed effects of ATR inhibition on nucleotide biosynthetic enzymes, RRM2 and dCK, are correlative and we have further expanded on this point in the revised manuscript.

6. *It would be interesting to compare the levels of the biosynthesis pathways in the cells of the dasatinib resistant pre-B ALL in the mice responding to treatment (13 of 20) vs the ones that don't respond (7 of 20). This experiment would help determine if there are other pathways at play or if there is another resistance mechanism that compensates for the biosynthesis pathway knockdown. This approach would provide important information for selecting patients who would gain a clinical benefit from this treatment (i.e. clinical biomarkers to determine which patients would benefit from this therapy).*

The reviewer raises an interesting point about clinical biomarkers to identify patients that would benefit from this therapy. At the time, we performed those experiments we were also concerned that the failure of 7 out of 20 mice to achieve complete remission may reflect acquired resistance to the combination therapy. We have carried out additional experiments to test this possibility. Dasatinib-resistant p185^{BCR-ABL} Arf^{-/-} pre-B ALL cells were harvested from the bone marrow of a moribund mouse who showed only a transient response to treatment. We then assayed the susceptibility of these cells to the combination therapy *ex vivo*. As shown in new Supplementary Fig. 14, the response of these cells to the combination therapy was indistinguishable from that of the parental cells, prior to inoculation in mice. A potential reason for the incomplete response in some of the treated mice is the rapid engraftment of p185^{BCR-ABL} Arf^{-/-} T315I+ pre-B-ALL cells in the brain coupled with the suboptimal penetrability of 3-AP and, potentially, dCKi across the blood-brain-barrier. We plan, in further studies, to search for biomarkers to stratify individuals for potential response.

Reviewer 1 also listed 13 Minor Points, all of which were addressed in the revised manuscript.

Reviewer #2

1. Fig. 1: The authors use $^{13}\text{C}_6$ -glucose to assess the effect of ATR inhibition on *de novo* nucleotide synthesis. The major issue of this experiment is that glucose is not only a substrate for the *de novo* pathways, it is also a substrate for the salvage pathway. Indeed, glucose can be converted through the pentose phosphate pathway into PRPP which is a substrate for *de novo* purine and pyrimidine synthesis but also for the salvage pathway (ie, APRT/HPRT). Assessing the incorporation of 13-carbon from ^{13}C -glucose into UTP, CTP and CDP does not give a read out of the activity of *de novo* nucleotide synthesis pathway. The authors should show other metabolites specific for the *de novo* pathway (ie, AICAR, Adenylosuccinate (*de novo* purine) and *N*-carbamoyl-aspartate, orotate, or orotidine monophosphate (*de novo* pyrimidine)). If it is not possible, the authors should perform the tracing experiment with another substrate specific for *de novo* nucleotide synthesis. Plus, the authors should measure glucose uptake.

The reviewer is correct – PRPP generated from glucose via the oxidative and/or non-oxidative pentose phosphate pathway can be used as a substrate for the nucleobase salvage pathways through the action of phosphoribosyltransferases (for instance HPRT and APRT for the purine nucleobase salvage pathway, and URPT for the pyrimidine nucleobase salvage pathway). Regarding purine biosynthesis, the contribution of these pathways to dATP and dGTP pools were investigated in detail in Supplementary Figures 5 and 6. Our data indicate that glucose-derived PRPP does indeed contribute significantly to nucleobase salvage derived dATP (Supplementary Fig. 5c) and dGTP (Supplementary Fig. 6c) incorporated into the newly replicated DNA of T-ALL cells. These contributions require the activity of adenosine deaminase (ADA) and purine nucleoside phosphorylase (PNP), which generate the substrates for the nucleobase salvage pathway (hypoxanthine and guanosine), as indicated by changes in dATP and dGTP biosynthesis induced by specific ADA and PNP inhibitors (Supplementary Fig. 5 and 6). In contrast to the purine nucleobase salvage pathway, in the analyzed T-ALL cells we did not observe evidence of active pyrimidine nucleobase salvage pathway which would utilize glucose labeled PRPP. These findings are consistent with data from the literature that in leukemia cells pyrimidines are salvaged as intact nucleosides, while purines are salvaged as nucleobases.

Regarding the effects of ATR inhibition on *de novo* pyrimidine biosynthesis, the *de novo* metabolite, orotidine monophosphate (OMP), was detected in our non-targeted LC-MS analysis (Fig. 1c); ATR inhibition decreased glucose labeling of OMP by 30%.

Regarding glucose uptake, we did not detect any changes in glycolytic intermediates (glucose, glucose 6-phosphate/fructose 6-phosphate, fructose 1,6-bisphosphate, dihydroacetone phosphate, glyceraldehyde 3-phosphate, 3-phosphoglycerate, phosphoenolpyruvate, pyruvate, and lactate) using our non-targeted LC-MS analysis (Supplementary Fig. 3). These data indicate that ATR inhibition does not interfere with glucose uptake.

*Fig.2: Incorporation of ^{15}N - ^{13}C from ^{15}N - ^{13}C -deoxycytidine (deoxynucleoside) into dCTP is specific for the salvage pathway since it is bypassing the need for glucose, however, other method should be used to assess the *de novo* pathway.*

As shown in Supplementary Figure 4 (and described in detail in Supplementary Information), our targeted MS assay distinguishes between the incorporation of glucose derived ^{13}C atoms in the nucleobase and in the ribose moiety. The ability to make this distinction makes it possible to simultaneously account for the use of glucose derived PRPP in nucleobase salvage pathways (as described above) and for the use of glucose in the *de novo* pathway (e.g. independent of nucleobase and/or nucleoside salvage pathways). To better illustrate these points and address the concerns of Reviewer 2 regarding the specificity of our targeted LC-MS/MS-MRM assay, we have included a new Supplementary Table 4 showing a detailed analysis of isotopologue composition in representative [$^{13}\text{C}_6$]glucose and [$^{13}\text{C}_9$, $^{15}\text{N}_3$]dC labeling experiment.

Fig.3: The phosphorylation of RRM2 on T33 is interesting. The authors should demonstrate whether ATR directly phosphorylate that site. The author should demonstrate whether the regulation of RRM2 by ATR is transcriptional or post-translational or both. Utilization of genetic approaches to confirm this regulation is needed (siRNA anti ATR or siRNA anti-dCK or combination).

We apologize for the incomplete explanation regarding this post-translational modification. RRM2 T33 is not phosphorylated directly by ATR. Instead, ATR indirectly regulates this site via CDK1/2. In the revised manuscript, we have modified the text to clearly state that RRM2 pT33 is a CDK-mediated phosphorylation event which promotes RRM2 proteasomal degradation via interactions with the SCF^{Cyclin F} ubiquitin ligase complex (ref #44, D'Angiolella et al., 2012). The text now reads as follows:

Results section, p7

“The reduction in RRM2 protein levels induced by ATR inhibition was accompanied by an ~50% decrease in the phosphorylation of RRM2 on threonine 33 (pT33) (Fig. 2e), a phosphosite previously linked to the stability of the RRM2 subunit⁴⁴”

Reviewer #3

1. Whilst ATR inhibition causes statistically significant decreases in nucleotide synthesis and salvage, it is only a small decrease, and the remaining levels are far more than 'residual'.

We agree with the Reviewer and we have revised the text accordingly.

1. Abstract p2, we replaced 'residual' with 'substantial remaining': "...ATR-inhibited acute lymphoblastic leukemia (ALL) cells revealed substantial remaining *de novo* and salvage activities which prevented replication shutdown...”

2. Results section p7, we replaced 'residual' with 'remaining': "To identify the most potent clinically relevant RNR inhibitors that could be used to target the remaining *de novo* biosynthetic activity in ATR inhibited CEM cells we evaluated four compounds...”

3. Results section p8, we replaced 'residual' and added 'remaining': "Having identified 3-AP as a clinically relevant and potent RNR inhibitor that can be used to directly target the remaining *de novo* activity in ATR treated T-ALL cells...”

4. Discussion p11, we replaced 'residual' with 'significant remaining': "However, significant remaining activities of both pathways were sufficient to prevent ATR inhibition-induced DNA replication shutdown in cell culture...”

2. It is not clear that the impact of ATR inhibition on cell survival is primarily due to nucleotide depletion. It is possible that the primary action of ATR inhibition is independent of nucleotide depletion, but that the combination of ATR inhibition and metabolic inhibition has an additive effect resulting in decreased cell survival. Some critical controls are not shown that would further support the claim of synthetic lethality. In particular, for the in vitro tests (fig 4b-c) the combination of metabolic inhibitors (D + AP) is not included. For the in vivo study, there is no comparison to single or dual inhibitor combinations to confirm that this triple-therapy is actually necessary.

We agree with Reviewer #3 that the impact on ATR inhibition on cell survival may not primarily due to nucleotide inhibition. In response to Reviewer #1's suggestion, we have attempted to rescue the effects of ATR inhibition on cell cycle kinetics by nucleotide supplementation. These experiments (new Supplementary Fig. 2b) indicate that nucleotide supplementation partially rescues the alterations in cell cycle kinetics caused by ATR inhibition, a result consistent with Reviewer #3's suggestion. The text was modified accordingly.

Results section p5

“Co-inhibition of ATR and dCK decreased the percentage of cells that reached S2 by five-fold relative to untreated cells (Fig. 1b). The effects of ATR inhibition on cell cycle kinetics were partially rescued by nucleotide supplementation, in a dCK-dependent manner (Supplementary Fig. 2c and 2d).”

Regarding the use of additional controls, in the revised manuscript we have – as suggested by the reviewer – examined the effects of all the possible combinations on cell viability (new Fig. 5e) and growth (Supplementary Fig 10) using p185^{BCR-ABL} *Arf*^{-/-} pre-B-ALL cells; we have also expanded the CEM T-ALL cell viability and phospho-Histone 3 data to include all possible combinations (Supplementary Fig 9a and 9b). Furthermore, as suggested by the Reviewer, we performed a new *in vivo* experiment to directly examine the efficacy of the two most potent treatment combinations (RNRi + ATRi) and (RNRi+ATRi+dCKi) in cell culture. As shown in Supplementary Fig. 12, omitting the dCK inhibitor from the combination therapy significantly reduced therapeutic efficacy, demonstrating the requirement for all three components for optimal therapeutic efficacy.

3. The metabolomics (untargeted metabolite profiling) experiment is not described in the methods. How were these samples extracted and analysed? Whilst a volcano plot is shown, only the nucleotides are labelled, whereas there are several metabolites that exhibit a larger change that are not labelled. If those metabolites are not nucleotides then it would argue against the primary mechanism of ATR inhibition involving nucleotide disruption (unlike the proteomics data that clearly shows RRM2 and TYMS and the most perturbed proteins). Furthermore, only the isotope enriched metabolites are shown. It would be interesting to see the full metabolomics data of relative metabolite levels to gauge the importance of nucleotide perturbation in the overall impact of ATR inhibition.

We thank the reviewer for raising the issue of the metabolomics (untargeted metabolite profiling) experiment not being described in the methods. We have included a detailed description of the methodology used for the untargeted metabolite profiling assay in the Supplementary Information section.

Regarding the actual data, 21 of the 44 [¹³C₆]glucose-labeled metabolites that were significantly altered in the treatment groups (FDR ≤ 20%) are involved in nucleotide metabolism. Of those, 19 metabolites show at least 15% change in glucose labeling in either ATR or combined ATR and dCK inhibition experimental groups. As suggested by the Reviewer, we have included the entire data set from the two independent untargeted LC-MS studies in Supplemental Table 1. The effects of ATR inhibition on nucleotide metabolism were confirmed by the more sensitive targeted LC-MS/MS-MRM assay. To address the potential functional significance of these effects, we have performed the nucleotide supplementation experiments suggested by Reviewer #1.

4. For the targeted metabolite labelling experiments it is convenient and probably appropriate to simplify the analysis to the three major species representing the unlabelled, de novo and salvage pathways. However, one would expect a number of other isotopologues to exist, e.g. label incorporation into the uracil moiety in de novo synthesis, or recombination of unlabelled (d)ribose with labelled cytosine (if there is active phosphorylase activity) for the salvage pathway. Could you provide an overview of the complete labelling pattern derived from these species in the supplementary data, and whether that changes after incubation with inhibitors?

We have addressed this suggestion by adding the new Supplementary Table 4. All three inhibitors (ATRi, dCKi, and RNRi), in all possible combinations, are included in this analysis. The various isotopologues are identified by the combined number of heavy isotopes in the deoxyribose moiety (¹³C) and the number of heavy isotopes in the nucleobase moiety (¹³C and/or ¹⁵N). We did indeed observe label incorporation into the uracil moiety via the de novo pathway, as predicted by the reviewer. Regarding phosphorylase activity, we have observed it for the purine biosynthesis (Supplementary Figure 5 and 6), but not for pyrimidine biosynthesis, consistent with the absence of uridine phosphorylase 1 and 2 expression in lymphoblastic leukemia cells.

5. For the glucose labelling experiment it would be interesting to see the % label incorporation for the species that come between glucose and the nucleotides in the metabolic pathways. At a minimum PRPP or ribose 5-phosphate should be shown to confirm that any change in label incorporation is due to the de novo nucleotide synthesis pathway rather than isotope dilution in central carbon pathways.

The glucose labelling experiment in Figure 1d now includes ribose 5-phosphate; this metabolite exhibits complete [¹³C₆]glucose-labeling across all treatment groups, indicating no isotope dilution in the central carbon pathway.

6. For the proteomics experiments it is not clear how the dimethyl labelling experiments were designed. The manuscript describes triplex labelling, but there are 4 sample groups. It mentions n=3, but according to fig 3a you'd end up with n=6 for the NT and V groups. Please clarify which data is included (and reasons for excluding data if this occurred). Furthermore, for labelling-based quantification experiments it is critical to switch the labelling to avoid labelling artefacts. Please describe how this was done (or complete this extra experiment if required). Also, how was the False Discovery Rate calculated for the statistical analysis of the proteomics data (it is mentioned in the results, but not in the methods).

Due to the nature of comparing 4 sample groups and being limited to 3 channels for stable isotope reductive amination labeling, we ran two independent multiplexed experiments with each independent experiment including the non-treated sample group as the bridge sample, in order to compare treated samples across the

experiments. Therefore, one multiplexed experiment contained non-treated, VE-822 treated, and VE-822 + dCKi treated sample groups and the other multiplexed experiment contained non-treated and dCKi treated sample groups. Each multiplexed experiment was performed in triplicate. We have not encountered labeling bias in any one channel and thus did not switch the isotopic dimethyl labels across replicate experiments. To calculate the false discovery rate (FDR) for the proteomics data, the MS/MS spectra were searched against both the Uniprot human FASTA database and a decoy database of the Uniprot human database which read from C-terminus to N-terminus; Percolator was used to filter the data at 1% FDR at both the peptide and protein level. This description of the FDR calculation has been updated in the Supplementary Information.

7. In the phosphoproteomics, the decreased level of pRRM2 is described, however, total levels of RRM2 were also down. Therefore, one would expect the pRRM2 levels to also decrease. Therefore this is not a PTM modification (i.e. the % of phosphorylated RRM2 is the same), just a decrease in total RRM2 abundance.

In our analysis, the displayed quantitation for RRM2 pT33 is normalized to the total RRM2 fold change (Fig. 2e). The mean fold change of 0.43 ± 0.1 for RRM2 pT33 was calculated by normalizing the observed 0.34-fold decrease for pT33 as measured in the phosphoproteomic nLC-MS/MS analysis to the observed 0.77-fold decrease in total RRM2 protein as measured in the proteomic nLC-MS/MS analysis. We have made this explicit in the figure legend and text. The figure legend text now reads as follows:

Figure legends, p 18

“(e) Relative level of RRM2 pT33 normalized to RRM2 protein level from **(d)**, in asynchronous CEM cells treated with VE-822 (1 μ M) for 12 h (mean \pm SD, n = 3, unpaired two-tailed Student’s t-test).”

8. Some technical details are missing from the methods: (i) how many cells were used per sample for the metabolomics and proteomics studies (ii) details for the non-targeted LCMS method (as mentioned above) and (iii) Please justify the normalisation strategy whereby the TIC at a specified retention time was used... if there was matrix effect at a specific retention time this might result in enhanced TIC but signal suppression, in which case this normalisation strategy would actually worsen the quantitative accuracy.

We thank the Reviewer for noticing these omissions and we have corrected them in the revised manuscript. Briefly, metabolites and proteins were extracted from 1×10^6 cells, the methodological details of the non-targeted LC-MS method are now included in the Supplementary Information section, and the metabolites were quantified by summing the isotopologues.

REVIEWERS' COMMENTS:

Reviewer #1 (Remarks to the Author):

All concerns have been addressed satisfactorily. It is a nice study. Good work!

Reviewer #2 (Remarks to the Author):

The manuscript has been significantly improved. The authors have addressed my concerns and questions.

Reviewer #3 (Remarks to the Author):

The authors have addressed all major concerns in this revision and I recommend that this detailed study is now suitable for publication.